# Visualizing molecules of functional *human* profilin

Morgan L Pimm[1], Xinbei Liu[1], Farzana Tuli[1], Jennifer Heritz[1], Ashley Lojko[1], Jessica L Henty-Ridilla[1,2]*

[1]Department of Biochemistry and Molecular Biology, SUNY Upstate Medical University, Syracuse, United States; [2]Department of Neuroscience and Physiology, SUNY Upstate Medical University, Syracuse, United States

**Abstract** Profilin-1 (PFN1) is a cytoskeletal protein that regulates the dynamics of actin and microtubule assembly. Thus, PFN1 is essential for the normal division, motility, and morphology of cells. Unfortunately, conventional fusion and direct labeling strategies compromise different facets of PFN1 function. As a consequence, the only methods used to determine known PFN1 functions have been indirect and often deduced in cell-free biochemical assays. We engineered and characterized two genetically encoded versions of tagged PFN1 that behave identical to each other and the tag-free protein. In biochemical assays purified proteins bind to phosphoinositide lipids, catalyze nucleotide exchange on actin monomers, stimulate formin-mediated actin filament assembly, and bound tubulin dimers ($k_D$ = 1.89 μM) to impact microtubule dynamics. In PFN1-deficient mammalian cells, Halo-PFN1 or mApple-PFN1 (mAp-PEN1) restored morphological and cytoskeletal functions. Titrations of self-labeling Halo-ligands were used to visualize molecules of PFN1. This approach combined with specific function-disrupting point-mutants (Y6D and R88E) revealed PFN1 bound to microtubules in live cells. Cells expressing the ALS-associated G118V disease variant did not associate with actin filaments or microtubules. Thus, these tagged PFN1s are reliable tools for studying the dynamic interactions of PFN1 with actin or microtubules in vitro as well as in important cell processes or disease-states.

*For correspondence:
ridillaj@upstate.edu

**Competing interest:** The authors declare that no competing interests exist.

## Editor's evaluation

The development and rigorous characterization of fully functional, fluorescently labeled versions of profilin will be useful to cell biologists and biochemists who study the cytoskeleton. For cell biologists, these tools will allow a better understanding of the cellular dynamics of profilin and its interactions with different components of the cytoskeleton (actin networks and microtubules). Fluorescent profilins will also be precious to understand the consequences of different mutations. For biochemists, these tools offer new possibilities to study profilin dynamics in bulk assays or by live imaging.

## Introduction

Profilin-1 (PFN1) is a small (~15 kDa) cytosolic protein that interacts with actin monomers, poly-*L*-proline (PLP) containing ligands, phosphoinositide (PIP) lipids, and microtubules to regulate many essential cell behaviors (*Davey and Moens, 2020*; *Pimm et al., 2020*; *Karlsson and Dráber, 2021*). It maintains cellular reserves of unassembled actin monomers, catalyzes nucleotide exchange, and sterically inhibits new filament formation (*Goldschmidt-Clermont et al., 1992*; *Kaiser et al., 1999*; *Wolven et al., 2000*; *Suarez et al., 2015*; *Pimm et al., 2020*; *Colombo et al., 2021*). In contrast, PFN1 can drastically stimulate actin assembly when paired with certain PLP-containing ligands (e.g. formins or Ena/VASP) (*Manchester, 1998*; *Romero et al., 2004*; *Kovar et al., 2006*; *Breitsprecher*

and Goode, 2013; Henty-Ridilla and Goode, 2015; Funk et al., 2019; Zweifel and Courtemanche, 2020). Competitive interactions between PFN1, actin monomers, and PIP lipids transmit cellular signals from the plasma membrane to distinct intracellular vesicles (Lassing and Lindberg, 1985; Davey and Moens, 2020). Furthermore, PFN1 directly impacts microtubule organization and polymerization (Nejedlá et al., 2021; Nejedla et al., 2016; Henty-Ridilla et al., 2017; Pimm and Henty-Ridilla, 2021). Notably, the only methods used to deduce PFN1 functions have been indirect, as standard labeling methods compromise different facets of PFN1 function.

Conventional protein tagging strategies using a GFP-derived fluorescent tag on the C- or N-terminus impede PFN1 interactions with PIP lipids or PLP-containing ligands (Wittenmayer et al., 2000; Davey and Moens, 2020; Pimm et al., 2020). Moving the fluorescent tag to a protruding loop in PFN1 restores these interactions, unfortunately compromising full actin-binding capacity (Nejedla et al., 2017; Karlsson and Dráber, 2021). Furthermore, a split PFN1 approach cannot be universally applied to all profilin isoforms or homologs. To complicate matters more, high concentrations (8–500 μM) make fluorescent PFN1 difficult to image in cells (Wittenmayer et al., 2000; Funk et al., 2019). A popular imaging approach to circumvent some of these issues employs anti-PFN1 antibodies to stain the detergent-extracted remnants of fixed cells (Henty-Ridilla et al., 2017; Nejedlá et al., 2021; Nejedla et al., 2017; DeCaprio and Kohl, 2020). This technique unambiguously localizes PFN1 to stable cytoskeletal structures (i.e. stress fibers, the microtubules, and nuclear components) in cells. However, it obscures the spatiotemporal details required to measure the flux or dynamics of cellular pools of PFN1.

To date no uniform tagging strategy has produced a tagged PFN1 that retains key functions and can be used in biochemical assays or diverse cellular situations. Here, we engineered and characterized two new versions of tagged PFN1 that behave identically to the tag-free version. To overcome the drawbacks of previous approaches a flexible linker at the N-terminus is coupled with GFP-derived or titratable self-labeling approaches (i.e. SNAP-, Halo-, or CLIP-tags). Purified mAp-PFN1 binds PIP lipids, catalyzes nucleotide exchange on actin monomers, stimulates formin-mediated actin filament assembly, and impacts microtubule dynamics equivalent to the tag-less PFN1. In PFN1-deficient cells, tagged versions of PFN1 restore protein concentration, cell morphology, and cytoskeleton-based phenotypes to endogenous levels. Using titrations of self-labeling ligands, PFN1 was detected in the cytoplasm, nucleus, and on actin filaments and microtubules in N2a, 3T3, and U2OS cells. The microtubule localization was shifted by expressing specific function-disrupting mutations in PFN1 (R88E, Y6D, and ALS-related variant G118V). We anticipate these tools will be useful in future biochemical

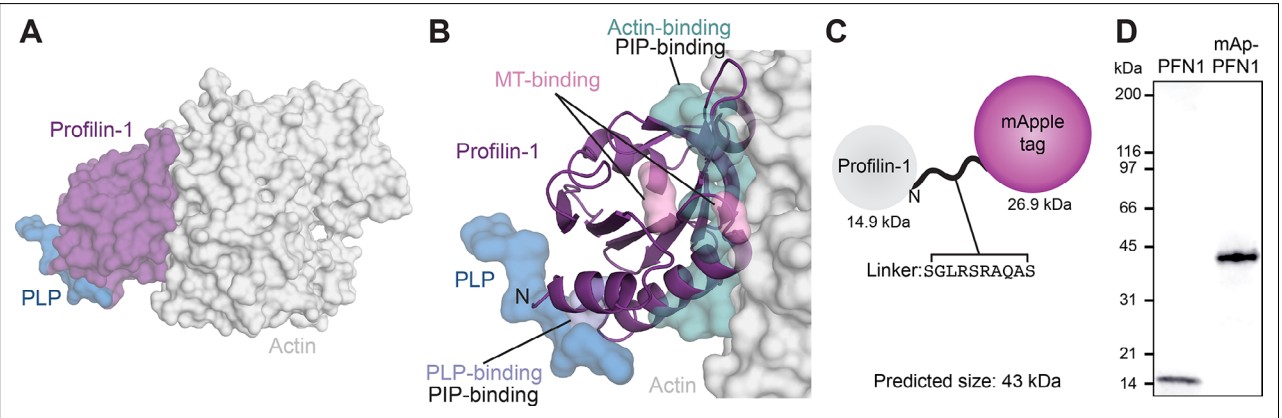

**Figure 1.** Strategy for fluorescently tagging and purifying profilin-1 (PFN1). (**A**) View of profilin (PFN1; purple) with an actin monomer (gray) and poly-*L*-proline (PLP; blue), from PDB: 2BTF and 2PAV. (**B**) Profilin residues that contact actin (teal), microtubules (MT) (M114 and G118; pink), PLP (Y6D; blue) or phosphoinositide (PIP) lipids. (**C**) Position of genetically encoded tag and linking sequence (SGLRSRAQAS) on PFN1. (**D**) Coomassie-stained SDS-PAGE gel of PFN1 and mApple-PFN1 (mAp-PFN1). Source file contains uncropped gel in (**D**).

The online version of this article includes the following source data and figure supplement(s) for figure 1:

**Source data 1.** Full SDS-PAGE gel and gel filtration trace.

**Figure supplement 1.** Gel filtration trace of mApple-profilin-1 (mAp-PFN1).

and cell-based studies exploring how PFN1 regulates the cytoskeleton to maintain homeostasis or trigger disease states.

## Results

### Design of tagged (PFN1)

Critical aspects of PFN1 function occur through conserved binding sites for PIP lipids, actin monomers, PLP motifs or microtubules (*Figure 1A and B*). To directly monitor PFN1 activities, we engineered two genetically encoded versions of the protein visible with either an mApple probe or self-labeling ligands that bind to Halo-tags. We chose the mApple probe over standard fluorescent proteins (e.g. GFP) because it is bright, relatively stable, and avoids overlap with the excitation and emission ranges of well-characterized actin-labels used in biochemical assays (e.g. Oregon-Green; OG, and pyrene-labeled actin; *Shaner et al., 2008*; *Bindels et al., 2017*). We fused each tag to the N-terminus of PFN1 flanked by a linking sequence, based on strategies used in the Michael Davidson Fluorescent Protein Collection (https://www.addgene.org/fluorescent-proteins/davidson/; *Figure 1C*), then expressed and purified recombinant versions of untagged PFN1 or mApple-PFN1 (mAp-PFN1) to compare the effects of the tag in several biochemistry assays. SDS-PAGE and gel filtration analyses revealed that the tag-free and tagged versions were of the expected size and highly pure (*Figure 1D* and *Figure 1—figure supplement 1*).

### mAp-PFN1 binds PIP lipids

PFN1 binds numerous PIP lipids through two different surfaces that overlap with either the actin- or PLP-binding sites (*Figure 1B*; *Hartwig et al., 1989*; *Lambrechts et al., 2002*; *Skare and Karlsson, 2002*; *Ferron et al., 2007*; *Nejedla et al., 2017*). PFN1-PI(4,5)P$_2$ interactions inhibit actin polymerization to regulate the morphology of the plasma membrane (*Niggli, 2005*; *Davey and Moens, 2020*). In addition, PFN1-PI(3,5)P$_2$ interactions facilitate trafficking of late endosomes to the lysosome (*Goldschmidt-Clermont et al., 1990*; *Martys et al., 1996*; *Dong et al., 2000*; *Hong et al., 2015*; *De Craene et al., 2017*; *Hasegawa et al., 2017*; *Wallroth and Haucke, 2018*; *Lees et al., 2020*). Thus, we performed liposome pelleting assays to assess whether mAp-PFN1 bound PI(3,5)P$_2$ or PI(4,5)P$_2$ lipids (*Figure 2A–C* and *Figure 2—figure supplement 1A, B*; *Banerjee and Kane, 2017*; *Chandra, 2019*). We detected mAp-PFN1 and untagged PFN1 bound to either PI(3,5)P$_2$- or PI(4,5)P$_2$-containing liposomes via Western blots using PFN1-specific antibodies (*Figure 2B-E*). Untagged

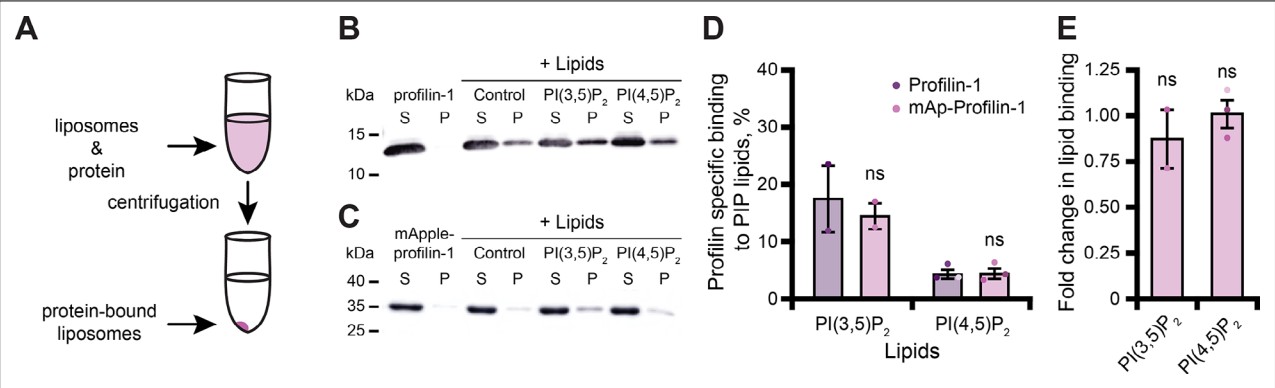

**Figure 2.** Tagged profilin binds phosphoinositide (PIP) lipids. (**A**) Schematic of liposome pelleting assay. (**B**) Blot of supernatant and pellet samples from 1 µM profilin-1 (PFN1) in buffer, PS control, or 0.33 mM PI(3,5)P$_2$ or PI(4,5)P$_2$. Blots were probed with anti-PFN1 primary antibody (1:5000; SantaCruz 137235, clone B-10) paired with goat anti-mouse:IRDye 800CW secondary (1:10,000; LI-COR Biosciences 926–32210). (**C**) Blot for 1 µM mApple-profilin-1 (mAp-PFN1). Assay as in (**B**). (**D**) Band densitometry after background subtraction. (**E**) Fold change in binding normalized to PFN1. Shaded values are independent data points (n=2–3). Error bars, SE. Not significant by Student's *t*-test, ns. Full blots in *Figure 2—figure supplement 1*. *Figure 2—source data 1* contains uncropped blots and quantification values for (**D and E**).

The online version of this article includes the following source data and figure supplement(s) for figure 2:

**Source data 1.** Full blots associated with PFN1 lipid binding experiments.

**Figure supplement 1.** Full blots associated with tagged profilin-1 (PFN1) binding phosphoinositide (PIP) lipids.

PFN1 was detected in pellets containing phosphatidylserine (control) or either PI(3,5)P$_2$ or PI(4,5)P$_2$ lipids (*Figure 2B-D*), but did not pellet without liposomes (buffer alone). Blot densitometry confirmed that mAp-PFN1 binds either PI(3,5)P$_2$ or PI(4,5)P$_2$ similar to tag-free PFN1 (*Figure 2D, E*).

## PFN1 and mAp-PFN1 are indistinguishable for actin-related functions

PFN1 is a well-characterized, high affinity (k$_D$ = 0.1 µM) actin monomer binding protein that catalyzes nucleotide exchange and suppresses actin filament nucleation (*Mockrin and Korn, 1980*; *Goldschmidt-Clermont et al., 1992*; *Eads et al., 1998*; *Vinson et al., 1998*; *Wolven et al., 2000*; *Wen et al., 2008*; *Blanchoin et al., 2014*; *Colombo et al., 2021*). To assess whether mAp-PFN1 binds actin monomers with similar affinity as untagged PFN1, we performed fluorescence polarization assays (*Figure 3* and *Figure 3—figure supplement 1*). Consistent with previous reports, we were unable to detect a change in polarization in reactions that contained untagged PFN1 and OG-labeled actin monomers (*Figure 3—figure supplement 1A*; *Vinson et al., 1998*; *Kaiser et al., 1999*). This is likely due to the short lifetime of the probe. However, several studies demonstrate that Thymosin β4 (Tβ4) and PFN1 compete to bind actin monomers (*Goldschmidt-Clermont et al., 1992*; *Aguda et al., 2006*; *Xue et al., 2014*). After confirming that GFP-Tβ4 bound actin (k$_D$ = 5.4 ± 1.2 nM; *Figure 3A*), we used competitive binding assays to measure the affinity of each PFN1 for actin monomers (*Figure 3B*). Each protein bound actin monomers with similar affinity (PFN1: k$_D$ = 99.6 ± 5.1 nM or mAp-PFN1: k$_D$ = 101.9 ± 6.5 nM; *Figure 3B*). These values are similar to those measured from direct binding assays (k$_D$ = 105.2 ± 26.9 nM; *Figure 3—figure supplement 1B*; *Wen et al., 2008*). To assess whether mAp-PFN1 is capable of catalyzing nucleotide exchange with similar efficiency as untagged PFN1, we performed time resolved fluorescence polarization assays containing ATP-ATTO-488, unlabeled actin monomers, and either PFN1 (*Colombo et al., 2021*). Each PFN1 catalyzed and accelerated the nucleotide exchange on actin monomers at similar rates (p=0.1; ANOVA) (PFN1: k = 18.7 ± 7.5 nM or mAp-PFN1: k = 13.2 ± 9.7 nM), but were accelerated compared to control reactions lacking either PFN1 (actin alone: k = 514.3 ± 61.6 nM) (*Figure 3C*).

Next, we assessed the ability of mAp-PFN1 to suppress spontaneous actin filament nucleation in bulk actin-fluorescence (pyrene) assays (*Figure 3D*). Less total actin polymer was made in the presence of either protein compared to the actin alone control (*Figure 3D*), demonstrating that both versions of PFN1 inhibit the spontaneous assembly of actin filaments. To examine this more carefully, we directly monitored actin filaments assembled in the presence of each PFN1 using total internal reflection fluorescence (TIRF) microscopy (*Figure 3E*). Reactions lacking PFN1 (actin alone), had an average of 45.3 ± 1.4 filaments per field of view (*Figure 3E and F*). In contrast, there were significantly fewer actin filaments in reactions supplemented with either PFN1 (21.5 ± 3.4 filaments) or mAp-PFN1 (19.8 ± 2.9 filaments; *Figure 3F*). As expected, neither protein substantially changed the mean elongation rate of actin filaments (control: 10.07 ± 0.16 subunits s$^{-1}$ µM$^{-1}$; PFN1: 10.07 ± 0.16 subunits s$^{-1}$ µM$^{-1}$; mAp-PFN1: 10.05 ± 0.16 subunits s$^{-1}$ µM$^{-1}$; *Figure 3G*). PFN1 is thought to bind to the fast-growing plus-ends of actin filaments at high concentrations (K$_D$ >20 µM; *Jégou et al., 2011*; *Courtemanche and Pollard, 2013*; *Pernier et al., 2016*; *Funk et al., 2019*; *Zweifel and Courtemanche, 2020*; *Zweifel et al., 2021*). Therefore, to assess whether or not mAp-PFN1 could be resolved on the end of actin filaments, we performed two-color TIRF microscopy at different actin-mAp-PFN1 stoichiometries (*Figure 3—figure supplement 2*). We observed few (<25 total) transient (>5 s) plus-end localization events (*Figure 3—figure supplement 2A-D*). In sum, these data demonstrate that mAp-PFN1 is equivalent to PFN1 for binding to actin monomers, catalyzing nucleotide exchange, and suppressing spontaneous actin assembly.

## PFN1 and mAp-PFN1 equally promote formin-based actin assembly

In addition to its role as a strong inhibitor of actin filament assembly (*Figure 3*), PFN1 can simultaneously bind actin monomers and proline-rich motifs (PLP) present in cytoskeletal regulatory proteins to stimulate actin filament assembly (*Kovar et al., 2006*; *Paul and Pollard, 2008*; *Breitsprecher and Goode, 2013*; *Zweifel and Courtemanche, 2020*). We used bulk pyrene fluorescence assays to assess whether mAp-PFN1 was capable of enhancing actin filament assembly through the PLP

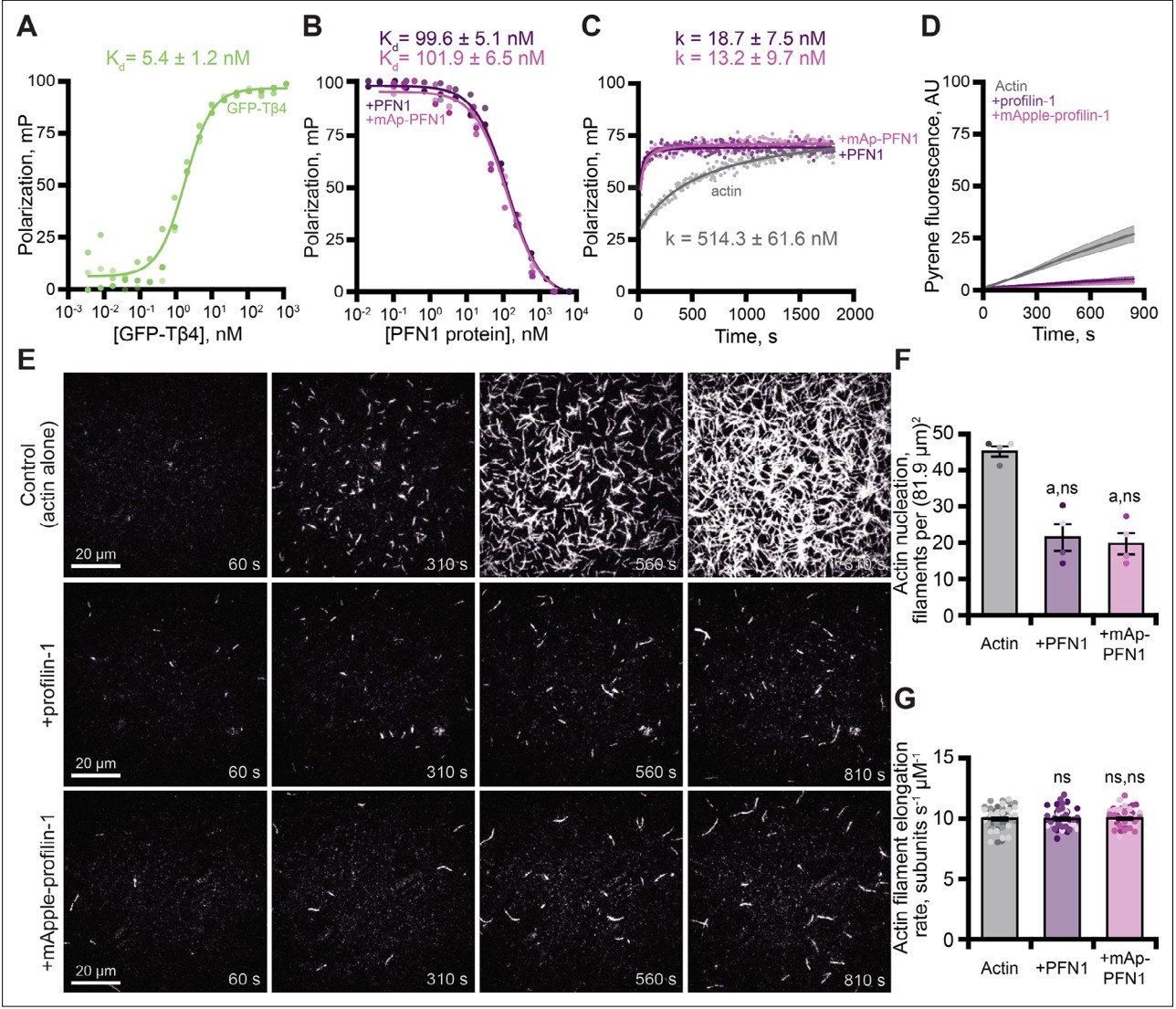

**Figure 3.** Tagged profilin binds actin monomers and stimulates nucleotide exchange. (**A**) Fluorescence polarization of 10 nM actin (unlabeled) with concentrations of GFP-thymosin β4 (GFP-Tβ4). (**B**) Competitive polarization of 10 nM GFP-Tβ4, 10 nM actin, and concentrations of profilin-1 (PFN1; purple) or mApple-PFN1 (mAp-PFN1; pink). (**C**) Kinetics of 500 nM ATP-ATTO-488–2 µM actin in the presence of MEI buffer (gray), 1 µM PFN1, or mAp-PFN1. Dots represent time resolved means from n=3 replicates. (**D**) Actin assembly via bulk fluorescence. Reactions contain: 2 µM actin (5% pyrene-labeled), and 3 µM PFN1 or mAp-PFN1. Shaded values are SE from n=3 assays. (**E**) Time lapse total internal reflection fluorescence (TIRF) of 1 µM actin (20% Oregon Green-labeled, 0.6 nM biotin-actin) in buffer (control), or 3 µM PFN1 or mAp-PFN1. Scale bars, 20 µm. See *Figure 3—video 1* and *Figure 4—video 2*. (**F**) Actin filament nucleation 100 s following initiation from movies as in (**E**), n=4 fields of view. (**G**) Actin filament elongation rates from movies as in (**E**) (n=51 filaments per condition). Shaded dots are single data points. Error bars, SE. Statistics, one-way ANOVA with Bartlett's correction: ns, not different; (**A**) p<0.05 from control. No difference found for mAp-PFN1 to PFN1. Source file contains quantification values for (**A–D and F and G**).

The online version of this article includes the following video, source data, and figure supplement(s) for figure 3:

**Source data 1.** Polarization readings for binding assays associataed with *Figure 3*, *Figure 3—figure supplement 1*.

**Figure supplement 1.** mApple-profilin-1 (mAp-PFN1) binds Oregon Green (OG)-actin monomers and is suitable for fluorescence-based binding assays.

**Figure supplement 2.** Localization of mApple-profilin-1 (mAp-PFN1) with actin filaments in vitro.

**Figure 3—video 1.** Total internal reflection fluorescence (TIRF) movie of profilin-1 (PFN1) or mApple-PFN1 (mAp-PFN1) on actin assembly.
https://elifesciences.org/articles/76485/figures#fig3video1

**Figure 3—video 2.** TIRF movie of mAp-PFN1 on actin assembly.Reaction contains: 1 µM actin (20% OG-labeled; 0.6 nM biotin-actin) (cyan) and 3 µM mAp-PFN1 (pink).
https://elifesciences.org/articles/76485/figures#fig3video2

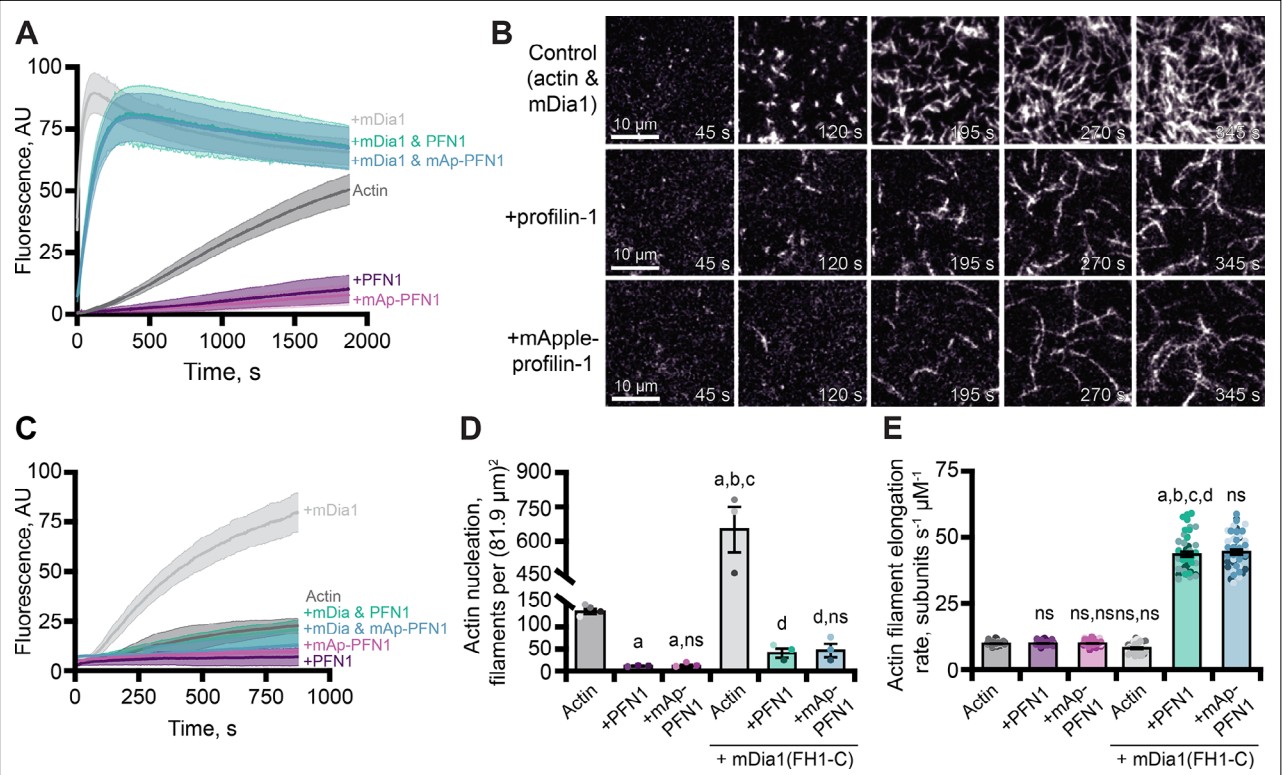

**Figure 4.** Effects of mApple-profilin-1 (mAp-PFN1) on formin-mediated actin assembly. (**A**) Bulk actin assembly: 2 μM actin (5% pyrene-labeled), 25 nM mDia1(FH1-C), and 5 μM profilin-1 (PFN1) or mAp-PFN1 (mAp-PFN1). Shaded values are SE from n=2 assays. (**B**) total internal reflection fluorescence (TIRF) of 1 μM actin (10% Alexa-647-labeled, 0.6 nM biotin-actin), 25 nM mDia1(FH1-C), and 5 μM PFN1 or mAp-PFN1. Scale bars, 10 μm. See *Figure 4—video 1*, *Figure 4—video 2* . (**C**) Actin fluorescence from TIRF videos. Shading indicates SE from n = 3 videos. (**D**) Mean nucleation 100 s after initiation, from n=4 fields of view. (**E**) Actin filament elongation rates from movies as in (**B**) (n=51 total filaments per condition). Shaded dots are individual data points. Error bars, SE. Statistics, one-way ANOVA with Bartlett's correction: ns, not different; (**A**) p<0.05 from control; (**B**) p<0.05 from PFN1; (**C**) p<0.05 from mAp-PFN1; (**D**) p<0.05 from actin and mDia1 control. No difference was found for mAp-PFN1 to PFN1. Source file contains quantification values for (**C–E**).

The online version of this article includes the following video, source data, and figure supplement(s) for figure 4:

**Source data 1.** Full views, nucleation, and elognation rate values for experiments exploring the effects of mAp-PFN1 in formin-based actin assembly assays.

**Figure supplement 1.** Full views of mApple-profilin-1 (mAp-PFN1) on formin-mediated actin assembly.

**Figure 4—video 1.** TIRF movie of PFN1 or mAp-PFN1 on formin-mediated actin assembly.
https://elifesciences.org/articles/76485/figures#fig4video1

**Figure 4—video 2.** Total internal reflection fluorescence (TIRF) video of mApple-profilin-1 (mAp-PFN1) in formin-mediated actin assembly assay.
https://elifesciences.org/articles/76485/figures#fig4video2

motif-containing formin, mDia1 (*Figure 4A*). Total actin filament polymerization reached similar levels in the presence of PFN1 or mAp-PFN1 (*Figures 3D and 4A*). Actin filaments assembled with the constitutively active formin mDia1(FH1-C) and either protein resulted in a strong enhancement in actin polymerization (*Figure 4A*). Using TIRF microscopy to directly monitor formin-based actin filament assembly we measured elevated levels of actin fluorescence from reactions that contained formin and either protein compared to controls (*Figure 4B–C* and *Figure 4—figure supplement 1A*). TIRF reactions containing actin and mDia1(FH1-C) had an average of 659.3 ± 98.3 filaments, which was significantly higher (p=0.014, ANOVA) than control reactions lacking formin or either PFN1 (*Figure 4D*). Consistent with previous reports that PFN1 also suppresses formin nucleation, reactions containing formin and either tag-free (42.7 ± 9.8 filaments) or mAp-PFN1 (48.3 ± 14.5 filaments) had statistically fewer filaments than control reactions containing formin and actin (p=0.0004, ANOVA) (*Kovar et al., 2006*; *Breitsprecher et al., 2012*; *Henty-Ridilla et al., 2016*).

To assess whether mAp-PFN1 was capable of stimulating formin-based actin assembly, we quantified actin filament elongation rates from TIRF movies (*Figure 4E* and *Figure 4—figure supplement 1B*). We measured actin filaments elongating at two different speeds that correspond to the recorded rates of unassisted growth (10.1 ± 0.4 subunits $s^{-1}$ $\mu M^{-1}$) and mDia1(FH1-C)-assisted growth in the presence of PFN1 (PFN1: 42.7 ± 9.8 subunits $s^{-1}$ $\mu M^{-1}$; or mAp-PFN1: 48.3 ± 14.5 subunits $s^{-1}$ $\mu M^{-1}$) (*Kovar et al., 2006*; *Henty-Ridilla et al., 2016*). No substantial change to the number of mAp-PFN1 end-association events in the presence was observed with formin in two-color TIRF assays. These data demonstrate that PFN1 and mAp-PFN1 stimulate formin-based actin assembly to comparable levels.

## PFN1 directly binds to tubulin dimers and enhances microtubule growth

In addition to functions with actin, PFN1 regulates microtubule polymerization (*Nejedla et al., 2016*; *Henty-Ridilla et al., 2017*). We used TIRF microscopy to monitor and compare the dynamics of microtubules polymerized in the presence of either protein (*Figure 5A* and *Figure 5—figure supplement 1A*). Microtubules in control reactions polymerized at an average rate of 1.9 ± 0.2 $\mu m^{-1}$ $min^{-1}$ (*Figure 5B–C*), whereas microtubules polymerized in the presence of either PFN1 grew at rates ~6 fold faster (PFN1: 12.7 ± 0.3 $\mu m^{-1}$ $min^{-1}$; mAp-PFN1: 12.6 ± 0.4 $\mu m^{-1}$ $min^{-1}$; *Figure 5B, C*; p<0.0001, ANOVA). Consistent with elevated microtubule growth rates, reactions performed in the presence of either PFN1 had significantly longer microtubules (*Figure 5D*; p<0.0001, ANOVA) that displayed more stable growth behaviors (*Figure 5E*; p<0.0001, ANOVA). Reactions containing either PFN1 had more microtubules than controls (*Figure 5F*; p=0.0048, ANOVA).

The combined regulatory activities of PFN1 on microtubule polymerization may be explained by several mechanisms, including that it binds or stabilizes tubulin dimers at growing microtubule plus-ends. To test for a direct interaction between mAp-PFN1 and unlabeled tubulin dimers, we performed fluorescence polarization ($k_D$ = 1.89 ± 0.28 $\mu M$) (*Figure 5G*). This affinity is 5.9-fold stronger than its affinity for polymerized microtubules ($k_D$ = ~10 $\mu M$) (*Henty-Ridilla et al., 2017*). We used TIRF microscopy to evaluate whether microtubule plus-ends were enriched with mAp-PFN1 (*Figure 5H–J* and *Figure 5—figure supplement 1B, C*). Rather than decorating plus-ends, mAp-PFN1 localized to the side of growing microtubules (*Figure 5H–I* and *Figure 5—figure supplement 1B*). Side-bound mAp-PFN1 molecules appear to diffuse along the microtubule lattice (*Figure 5J* and *Figure 5—figure supplement 1C*). Thus, mAp-PFN1 and PFN1 are equivalent regulators of microtubule dynamics, and promote assembly by direct interactions with dimers and the microtubule lattice.

## mAp-PFN1 restores endogenous protein levels and cell functions

Mutations in PFN1 are associated with neurodegenerative disease, including amyotrophic lateral sclerosis (ALS; *Wu et al., 2012*; *Murk et al., 2021*). Therefore, we generated and confirmed two clonal CRISPR/Cas9 knockout lines (PFN1$^{(-/-)}$) in immortalized neuroblastoma-2a (N2a) cells (*Figure 6A* and *Figure 6—figure supplement 1A-C*) that have been successfully used to model features of ALS (*De Vos, 2007*; *Vance et al., 2009*; *Coussee et al., 2011*; *Henty-Ridilla et al., 2017*). PFN1 deficient cell lines did not proliferate as efficiently as endogenous controls (*Figure 6B*), consistent with previously reported cell cycle defects (*Suetsugu et al., 1998*; *Witke et al., 2001*; *Moens and Coumans, 2015*). We used quantitative Western blots to evaluate whether PFN1 or tagged proteins (mAp-PFN1 or Halo-PFN1) restored PFN1 protein to endogenous levels (*Figure 6C* and *Figure 6—figure supplement 1D-F*). N2a cells contain 121 ± 15 $\mu M$ PFN1 and each plasmid harboring tagged or tag-free PFN1 restored the protein to endogenous levels (*Figure 6D*).

To assess whether mAp-PFN1 also restored cell-based functions, we plated cells on micropatterns to standardize the shape and measured cell morphology (*Figure 6E and F* and *Figure 6—figure supplement 1G*), the fluorescence of actin filaments (*Figure 6G and H*) or the fluorescence of microtubules (*Figure 6I and J*) in endogenous, knockout, and PFN1$^{(-/-)}$ cells transfected with tag-free or mAp-PFN1 plasmids. Notably, traditional assays for assessing the cytoskeleton in these cells are difficult to perform because N2a cells do not efficiently migrate or divide. PFN1$^{(-/-)}$ knockout cell morphology significantly deviated from all cells expressing PFN1 (endogenous, tagged, or untagged; p=0.01; ANOVA; *Figure 6E and F*). In addition, knockout cells displayed strikingly aberrant actin filament (p=0.02; ANOVA; *Figure 6G and H*) and microtubule networks (p=0.002; ANOVA; *Figure 6I and J*). Compared to PFN1$^{(+/+)}$, the cell morphology, architecture of actin filaments, and shape of the

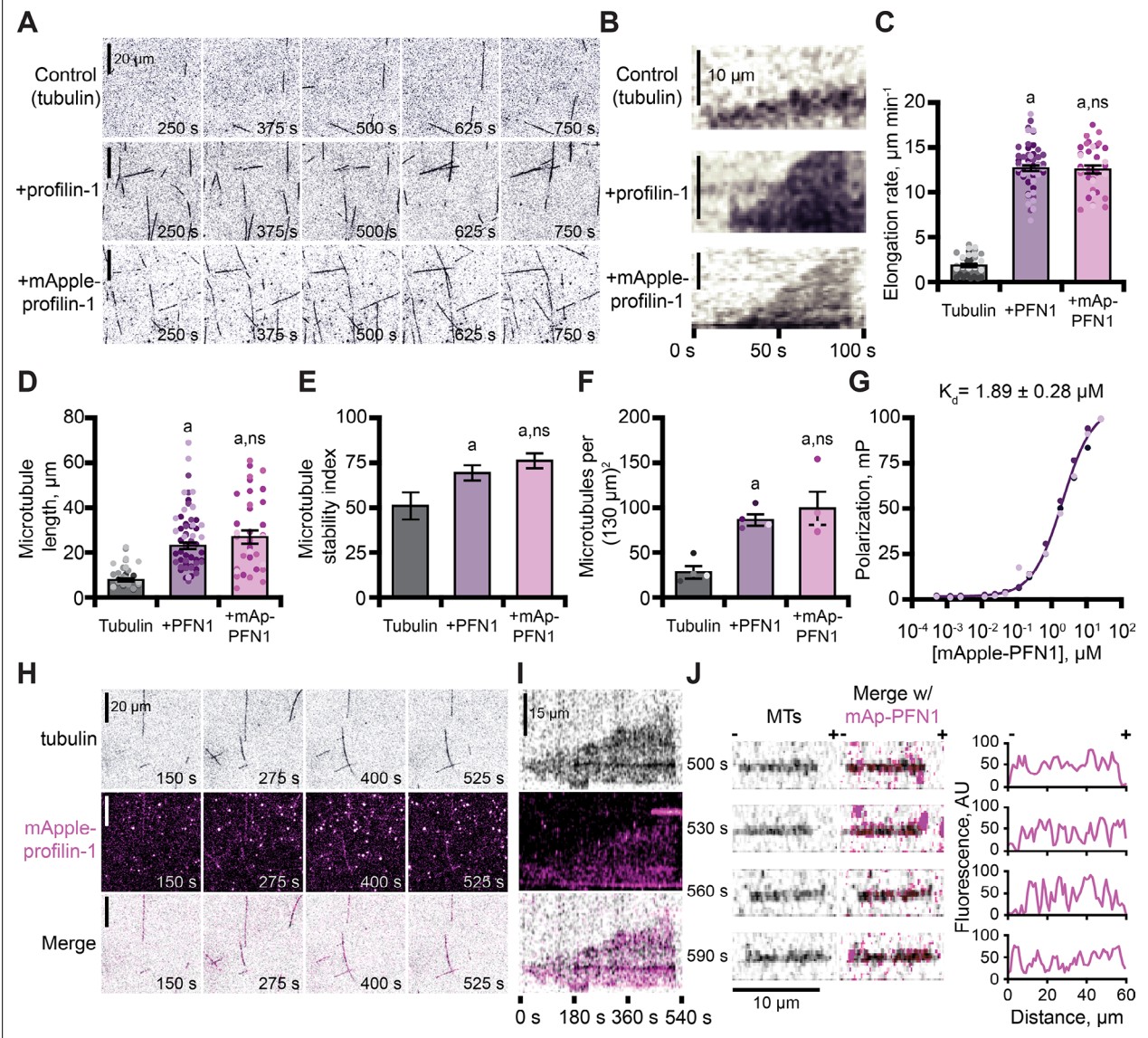

**Figure 5.** Profilin binds tubulin dimers and associates with the microtubule lattice. (**A**) Total internal reflection fluorescence (TIRF) images of reactions of biotinylated GMP-CPP seeds, 10 µM free tubulin (5% HiLyte-488 labeled), and buffer (control) or 5 µM profilin-1 (PFN1) or mApple-PFN1 (mAp-PFN1). Scale bars, 20 µm. See *Figure 5—video 1*. (**B**) Kymographs display dynamics as in (**A**). Scale bars: length, 10 µm; time, 100 s. (**C**) Microtubule polymerization (35–58 microtubules from n=3 experiments). (**D**) Microtubule length (n=35–58 microtubules from n=3 experiments). (**E**) Stability index: rescue/catastrophe frequency (n=18–46 microtubules from n=3 experiments). (**F**) Number of microtubules (n=4 experiments). Shaded dots are single data points. Error bars, SE. (**G**) Polarization of 10 nM tubulin (unlabeled) and mAp-PFN1. (**H**) TIRF as in (**A**), but visualizing tubulin (black) and mAp-PFN1 (pink). Scale bars, 20 µm. See *Figure 5—video 2*. (**I**) Kymographs display dynamics in (**H**). Scale bars: length, 15 µm; time, 540 s. (**J**) Montage of a microtubule (black) and mAp-PFN1 (pink) merged, with the intensity profiles of mAp-PFN1 along the microtubule lattice. See *Figure 5—video 3*. Scale bar, 10 µm. + and -, microtubule polarity. Statistics, one-way ANOVA with Bartlett's correction: (**A**) p<0.05 from control. No difference was found for mAp-PFN1 to PFN1. Source file contains quantification values for (**C–G and J**).

The online version of this article includes the following video, source data, and figure supplement(s) for figure 5:

**Source data 1.** Full views of TIRF movies and values for measured parameters associated with *Figure 5*.

**Figure supplement 1.** Additional views of the effects of mApple-profilin-1 (mAp-PFN1) on microtubule dynamics.

**Figure 5—video 1.** TIRF movie of PFN1 or mAp-PFN1 on microtubules.

https://elifesciences.org/articles/76485/figures#fig5video1

**Figure 5—video 2.** TIRF movie of mAp-PFN1 on microtubules.

https://elifesciences.org/articles/76485/figures#fig5video2

Figure 5 continued

**Figure 5—video 3.** mAp-PFN1 transiently associates with the microtubule lattice.

https://elifesciences.org/articles/76485/figures#fig5video3

microtubule network was not different than PFN1$^{(-/-)}$ cells expressing tag-free or mAp-PFN1 plasmids (p=0.78; ANOVA; *Figure 6E and J* and *Figure 6—figure supplement 1G*). This demonstrates that mAp-PFN1 complements cytoskeleton-based activities in cells.

## Aberrant PFN1 regulation of actin filaments and microtubules contributes to ALS

Eight mutations in PFN1 cause ALS (*Wu et al., 2012*; *Liu et al., 2022*). However, the detailed mechanisms that explain how PFN1 variants contribute to the disease state remain unclear. Empowered by the knowledge that mAp-PFN1 fully compensates for PFN1, we measured cell and cytoskeletal morphology parameters for PFN1$^{(-/-)}$ cells expressing Halo-PFN1(G118V), an ALS variant that does not effectively bind actin monomers or microtubules, (*Figure 7*; *Henty-Ridilla et al., 2017*; *Liu et al., 2022*). Consistent with analyses comparing cells expressing mAp-PFN1 (*Figure 6E–J*), PFN1 deficient cells displayed aberrant cell morphology (p=0.02; ANOVA; *Figure 7A and B*), as well as significantly decreased fluorescence of actin filaments (p=0.03; ANOVA; *Figure 7C and D*) and microtubules (p=0.01; ANOVA; *Figure 7E and F*). PFN1$^{(-/-)}$ cells expressing only Halo-PFN1(G118V) were unable to compensate for any of these parameters (p=0.01; ANOVA; *Figure 7A–F*). Thus, PFN1(G118V) contributes to ALS through actin and microtubule-based mechanisms.

## Live-cell visualization of Halo-PFN1

The discrete cellular localization of PFN1 has been difficult to determine because the endogenous protein is present at high cellular concentrations (121 µM; *Figure 6D*; *Pollard et al., 2000*; *Funk et al., 2019*; *Skruber et al., 2020*). Therefore, to further confirm the utility of the fluorescently-tagged protein, we performed live-cell imaging of PFN1 deficient N2a cells transfected with Halo-PFN1 and markers for actin or microtubules (*Figure 8*, *Figure 8—figure supplements 1 and 2*). We performed a titration of Janelia Fluor-646 (JF-646) ligand to limit the amount of cellular PFN1 signal and to test whether molecules of the protein could be visualized coincident with actin or microtubules (*Figure 8A*). At high concentrations of JF-646, Halo-PFN1 was localized to the nucleus and throughout the cytoplasm (*Figure 8A*). Single molecules of Halo-PFN1 were identified at 1–10 nM ligand, and localized to microtubules in ~60% of N2a cells (*Figure 8A–C*).

To explore whether this localization was cell-type specific, we overexpressed Halo-PFN1 in additional mammalian cell types (*Figure 8—figure supplement 1A*). In NIH 3T3 cells, Halo-PFN1 localized to actin filaments in all cells assessed and only occurred with microtubules at sites where both polymers overlapped (*Figure 8—figure supplement 1A, B*). In contrast, Halo-PFN1 associated more frequently with microtubules than actin filaments for U2OS and N2a cells (*Figure 8—figure supplement 1A-D*). Notably, for N2a cells, the localization of Halo-PFN1 to actin filaments only occurred at sites where both polymers overlapped (*Figure 8—figure supplement 1D*). Expressing Halo-PFN1 over endogenous PFN1 did not stimulate further microtubule association (*Figure 8—figure supplement 1E*).

To shift cellular PFN1 pools from the endogenous localization, we transfected PFN1$^{(-/-)}$ cells with Halo-PFN1 plasmids harboring point mutations that disrupt specific PFN1 functions (*Ezezika et al., 2009*; *Rotty et al., 2015*; *Suarez et al., 2015*; *Henty-Ridilla et al., 2017*; *Liu et al., 2022*). PFN1 colocalized with microtubules >50% of cells expressing the actin monomer binding deficient PFN1, Halo-PFN1(R88E) (n>50 cells from 5 separate experiments; *Figure 8B–C*). Line scans projected as kymographs further demonstrate that at least some protein is associated with microtubules (*Figure 8D* and *Figure 8—figure supplement 2*). However, it is still difficult to resolve whether these molecules are transiently associated or move along the microtubule lattice *Figure 8—video 1*. Surprisingly, PFN1 colocalized with microtubules in >75% of cells expressing the formin/poly-*L*-proline binding deficient Y6D mutation (*Figure 8B and C*; *Ezezika et al., 2009*; *Henty-Ridilla et al., 2017*; *Liu et al., 2022*). Microtubules in these cells appear less dynamic than wild-type or R88E-expressing cells, which may strengthen the hypothesis that PFN1-microtubule interactions stabilize microtubule dynamics

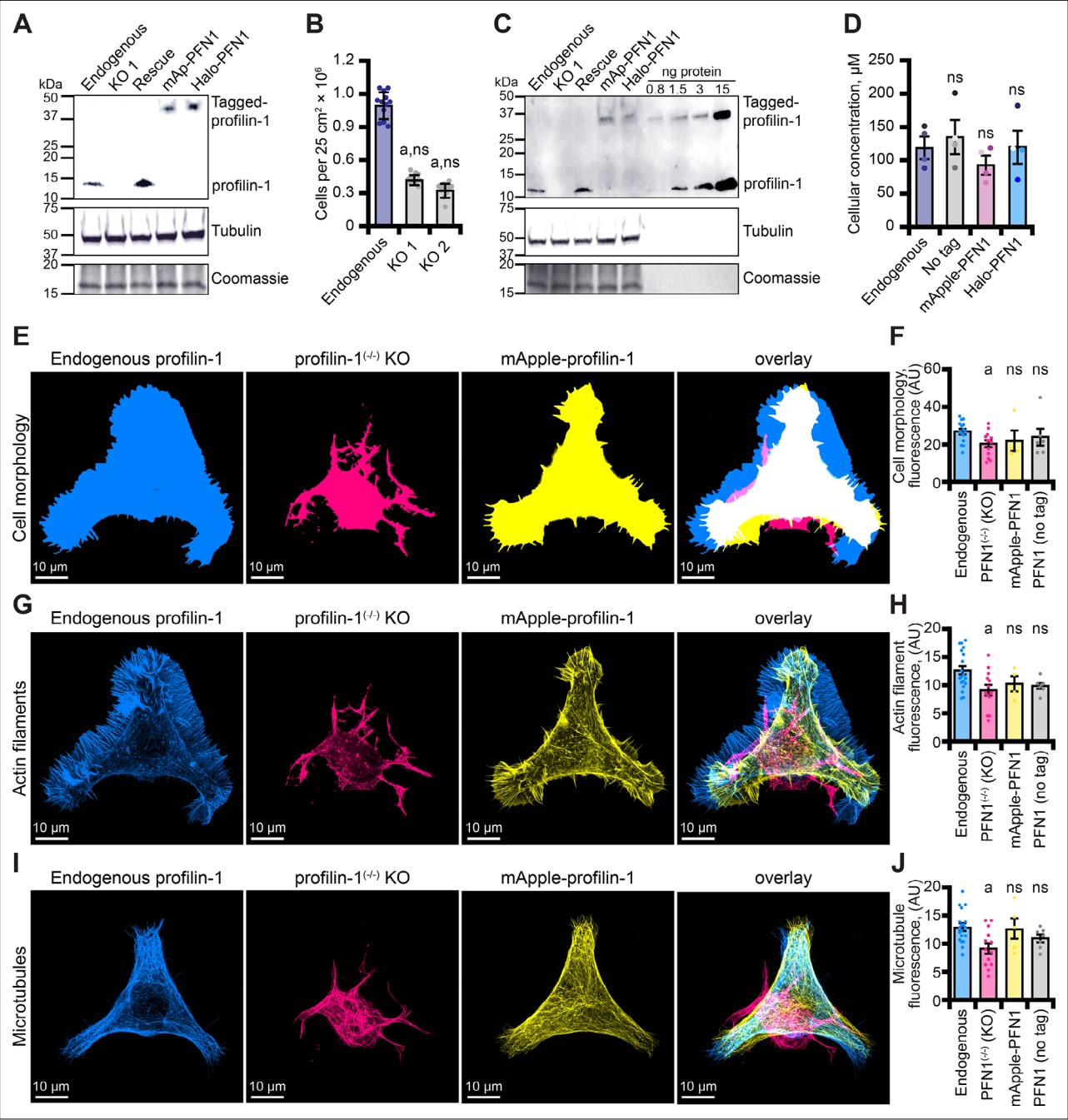

**Figure 6.** Tagged versions of profilin rescue protein levels and cell-based phenotypes in profilin-1 (PFN1) deficient cells. (**A**) Blot confirming knockout and rescue of PFN1 with tag-free PFN1, mApple-PFN1 (mAp-PFN1) or Halo-PFN1 plasmids. Extracts prepared from endogenous PFN1[(+/+)], PFN1[(-/-)], and PFN1[(-/-)] transfected with tag-free (rescue), mAp-PFN1 or Halo-PFN1 plasmids. Blot probed with anti-PFN1 antibody (1:3500; SantaCruz 137235, clone B-10) paired with goat anti-mouse:IRDye 800CW secondary (1:5000; LI-COR Biosciences 926–32210), and anti-α-tubulin (1:10,000; Abcam 18251) paired with donkey anti-rabbit 926–68073 secondary (1:20,000). Coomassie stained membrane used as loading control. Full blot in *Figure 6—figure supplement 1A-C*. (**B**) PFN1[(+/+)] cells proliferate significantly better than PFN1-deficient cell lines. (**C**) Blot probed as in (**A**) with known quantities of purified PFN1 and mAp-PFN1. Full blot in *Figure 6—figure supplement 1D-F*. (**D**) PFN1 levels in neuroblastoma (N2a) cells. Shaded dots are data points from n=4 experiments. Error bars, SE. Maximum intensity images and quantification of (**E–F**) cell morphology and (**G–H**) fluorescence of phalloidin-stained actin filaments or (**I–J**) microtubules from N2a cells expressing endogenous PFN1 (blue), PFN1[(-/-)] (pink), PFN1[(-/-)] transfected with mAp-PFN1 (yellow) or tag-free PFN1 plasmids (*Figure 6—figure supplement 1G*). Cells were plated on micropatterns and stained with anti-α-tubulin antibody (1:100; Abcam 18251) paired with donkey anti-rabbit conjugated to AlexaFluor-647 (1:100; Life Technologies A31573). Scale bars, 10 μm. Shaded dots are individual cells (n=4–15 cells) from n=3 coverslips. Error bars, SE. Statistics, one-way ANOVA with Bartlett's correction: ns, not different

*Figure 6 continued on next page*

*Figure 6 continued*

from PFN1$^{(+/+)}$; (**A**) p<0.05 from PFN1$^{(+/+)}$. No significant difference was found for mAp-PFN1, Halo-PFN1 or tag-less PFN1 to endogenous PFN1$^{(+/+)}$. Source file contains uncropped blots and quantification values for (**B, D, F, H,** and **J**).

The online version of this article includes the following source data and figure supplement(s) for figure 6:

**Source data 1.** Full blots and additional cell views associated with *Figure 6*.

**Figure supplement 1.** Full blots used to determine profilin-1 (PFN1) levels in neuroblastoma-2a (N2a) cells.

(*Figure 8D* and *Figure 8—video 1*). Significantly less PFN1 was associated with microtubules in cells expressing the microtubule-binding deficient, and ALS-relevant, G118V mutation (*Figure 8B and C*). These cells displayed PFN1 accumulations may be disease-relevant aggregates that are hypothesized to contribute to ALS onset (*Wu et al., 2012*; *Figley et al., 2014*). A line scan placed over a Halo-PFN1(G118V) expressing cell does not show much colocalization with microtubules (*Figure 8D*, *Figure 8—figure supplement 2*, and *Figure 8—video 1*). Thus, the localization of molecules of PFN1 can be shifted from actin- or microtubule-binding activities with specific point mutations.

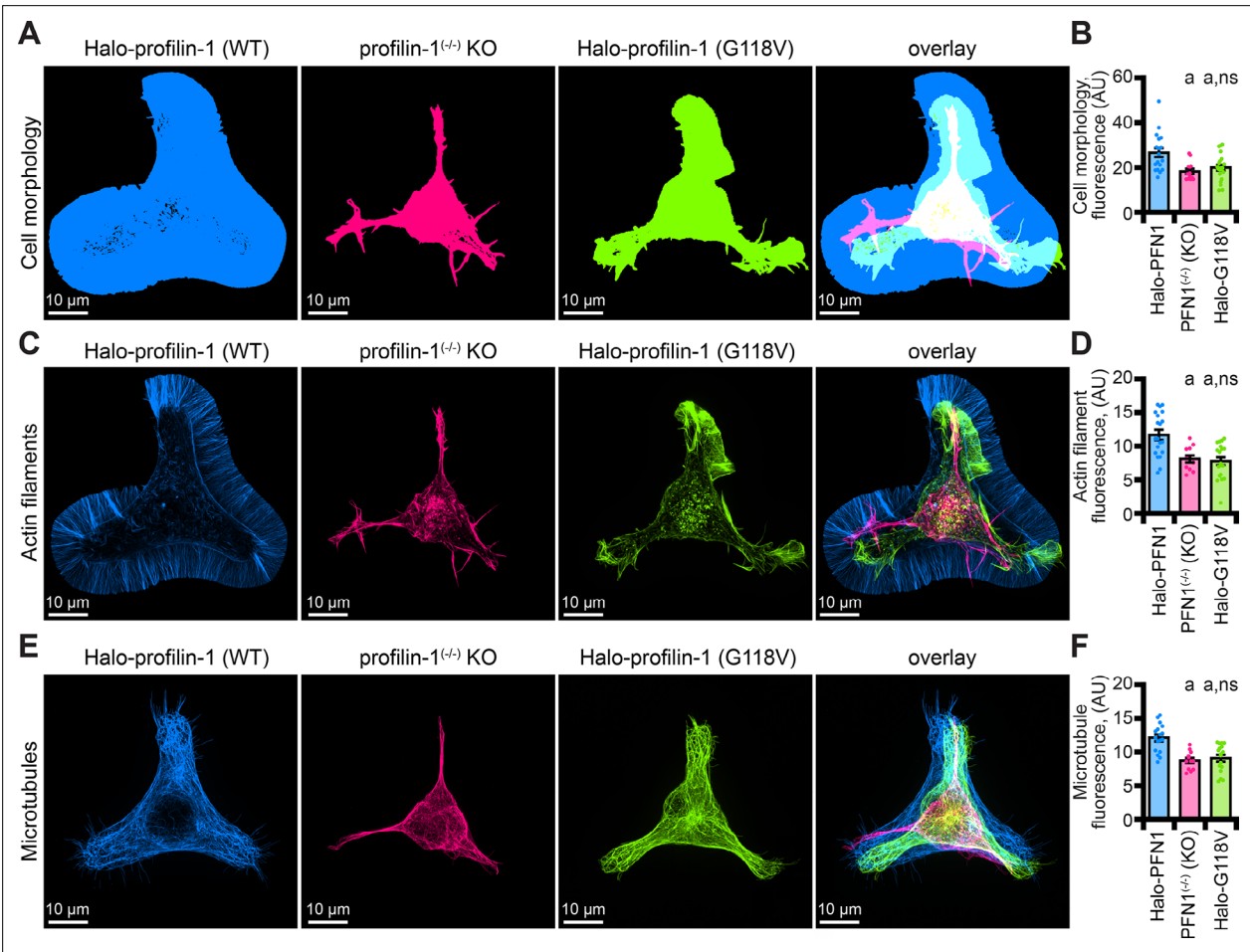

**Figure 7.** The profilin-1(G118V) amyotrophic lateral sclerosis (ALS) variant does not rescue morphology or cytoskeletal phenotypes present in profilin-1 (PFN1) deficient cells. (**A–B**) Maximum intensity images and quantification of cell morphology and (**C–D**) fluorescence of phalloidin-stained actin filaments (**E–F**) or microtubules from neuroblastoma-2a PFN1$^{(-/-)}$ transfected with Halo-PFN1 (blue), PFN1$^{(-/-)}$ (pink), or PFN1$^{(-/-)}$ cells transfected with Halo-PFN1(G118V; lime). Cells plated and stained as in *Figure 6I–J*. Scale bars, 10 μm. Shaded dots are individual cells (n=16–25 cells) from at least n=3 coverslips. Error bars, SE. Statistics, one-way ANOVA with Bartlett's correction: ns, not different from PFN1$^{(-/-)}$ expressing Halo-PFN1 (control); (**A**) p<0.05 from control. Source file contains quantification values for (**B, D,** and **F**).

The online version of this article includes the following source data for figure 7:

**Source data 1.** Source values for cell morphology, and total fluorescence of actin filaments and microtubules.

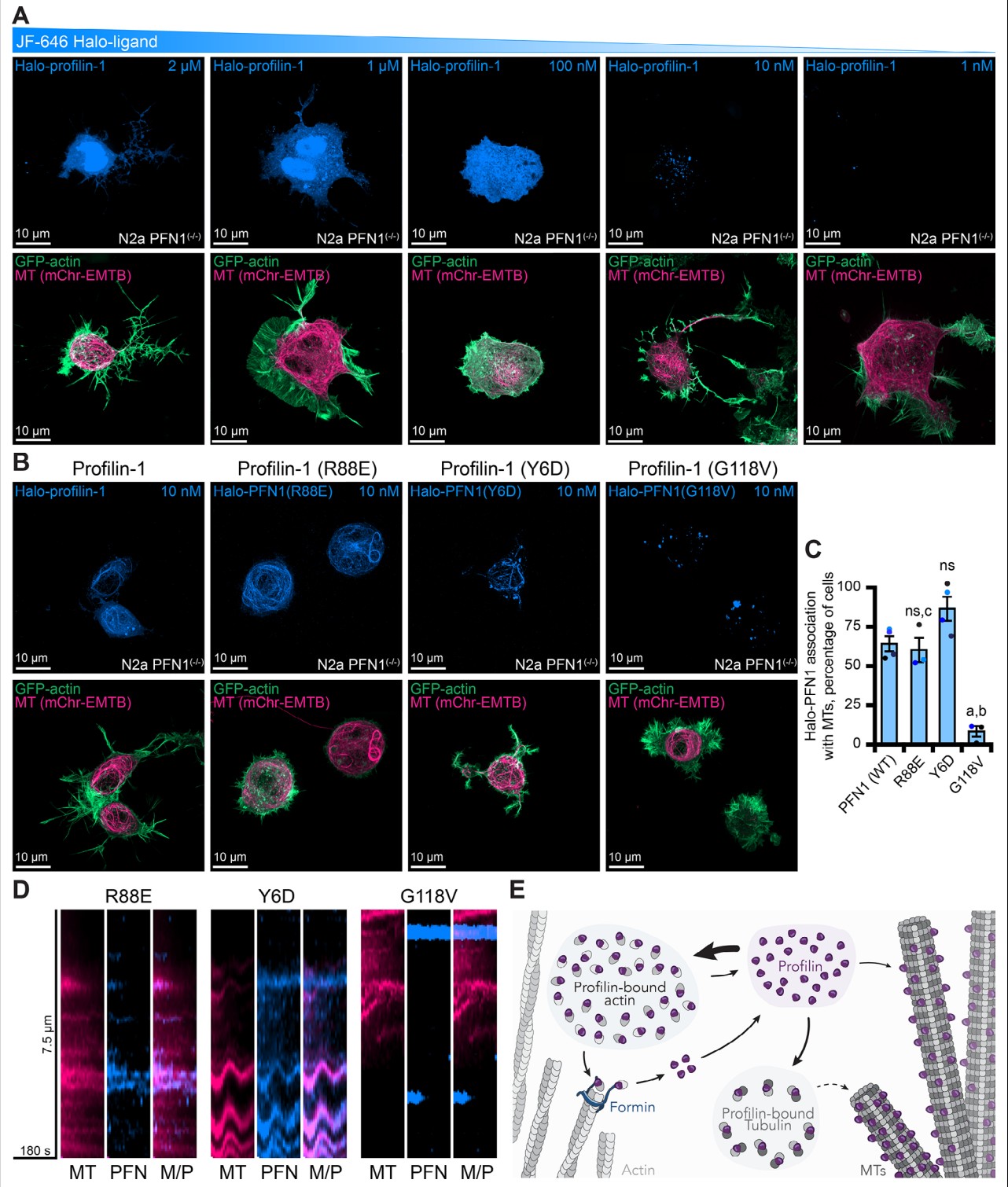

**Figure 8.** Live-cell visualization of individual molecules of profilin-1 (PFN1). (**A**) Maximum intensity projections of PFN1[(-/-)] expressing markers for actin (green), microtubules (pink) or Halo-PFN1 (blue; top) visualized with JF-646 and imaged as in *Figure 6*. Titration of JF-646 to illuminate PFN1 molecules. (**B**) PFN1[(-/-)] transfected with Halo-PFN1 plasmids (wild-type, R88E, Y6D, and G118V) as in (**A**). Scale bars, 10 µm. (**C**) Halo-PFN1-microtubule overlap in cells from (**B**). Overlap analysis was performed in n=3–4 experiments. Averages from 10 to 25 cells per experiment. Statistics, one-way ANOVA with Bartlett's correction: ns, not different from PFN1[(-/-)] expressing Halo-PFN1 (control); (**A**) p<0.05 from control; (**B**) p<0.05 from Halo-PFN1(R88E) or Halo-PFN1(Y6D); (**C**) p<0.05 from Halo-PFN1(G118V). (**D**) Kymographs display microtubule dynamics (see *Figure 8—figure supplement 2* and *Figure 8—video 1*). (**E**) Model of PFN1 distribution in cells. Source file contains quantification values for (**C**).

*Figure 8 continued on next page*

*Figure 8 continued*

The online version of this article includes the following video, source data, and figure supplement(s) for figure 8:

**Source data 1.** Halo-profilin microtubule localization counts.

**Figure supplement 1.** Localization of Halo-profilin-1 (Halo-PFN1) in different cell types.

**Figure supplement 2.** Live cell localization of constructs of Halo-profilin-1 (Halo-PFN1).

**Figure 8—video 1.** Halo-profilin-1 (Halo-PFN1) dynamics in live neuroblastoma-2aN2a cells.

https://elifesciences.org/articles/76485/figures#fig8video1

## Discussion

The lipid, actin, and microtubule regulating capabilities of profilin (PFN1) position it as a critical convergence point for major cell signaling pathways. Here we engineered and characterized genetically encoded tagged PFN1 proteins that are fully functional for interactions with important signaling lipids, binding and exchanging nucleotides on actin monomers, stimulating mDia1-based actin filament assembly, and binding to microtubules (*Figure 8E*). In cells, Halo-PFN1 fully compensates for the loss of the endogenous protein and restores normal cell morphology and actin filament and microtubule array architectures. A titration of specific self-labeling JF-646 ligand allowed us to directly localize a subset of PFN1 molecules for the first time in living cells. Overall, these tools directly illuminate functions of PFN1 that could not be deduced previously through indirect mechanisms or observed due to high cellular concentrations.

Fluorescent PFN1 also offers many advantages for dissecting direct PFN1-binding relationships. First, mAp-PFN1 is well-suited for fluorescence polarization assays to determine the affinities of PFN1 for binding actin monomers or tubulin dimers. This assay may now be expanded to disease-relevant mutations in PFN1 or to determine the binding constants of canonical ligands (i.e. formin, VASP, PLP, and PIPs) (*Liu et al., 2022*). Second, PFN1 was tagged with two entirely different genetically encodable probes through the same flexible linker. This validates that the tagging strategy may be applied with new fluorophores of different wavelengths or sizes (*Oliinyk et al., 2019*). Furthermore, when paired with appropriate and well-placed fluorophores, functional fluorescently tagged PFN1 may now be combined with FRET. Such studies paired with elegantly engineered formins may elucidate more details of how PFN1 interacts with PLP-tracts of different lengths or compositions at nm resolution (*Courtemanche and Pollard, 2013*; *Aydin et al., 2018*; *Zweifel and Courtemanche, 2020*).

Perhaps the greatest benefit of these tools is the ability to decipher mechanisms of PFN1 function with actin or microtubules in important cell processes. Dye-ligand titrations to illuminate Halo-PFN1 effectively overcome many of the problems associated with visualizing highly concentrated fluorescent proteins in cells. While advantageous, it is worth reiterating that only a fraction of the total cellular PFN1 is visible with this approach. As a consequence, Halo-PFN1 is the most appropriate for deciphering high affinity cellular mechanisms that occur at high concentrations and discrete cellular locations. Specifically, Halo-PFN1 is suitable for detecting interactions with microtubules, vesicles, stress fibers, and centrosomes rather than with actin monomers in the cytoplasm. However, the utility of these probes in the cytoplasm or at the leading edge may be achieved with optimization, genetics, different cell types, or clever techniques that stimulate PFN1 (*Lee et al., 2013*; *Skruber et al., 2020*).

Finally, several disease-variants of PFN1 have been identified in cancer and neurodegenerative disorders (*Michaelsen-Preusse et al., 2016*; *Pimm et al., 2020*; *Murk et al., 2021*). Using a genetic approach, we observed less microtubule association in cells expressing the actin- and microtubule-binding deficient PFN1(G118V) ALS variant. This is the first examination of the effects of an ALS-related PFN1 at endogenous levels. Future studies will detail how other functional or disease-related mutations influence the role of PFN1 in diverse signaling schemes, associated disorders, and different cell types.

## Materials and methods
### Reagents

All materials were obtained from Fisher Scientific (Waltham, MA) unless otherwise noted.

## Plasmid construction

DNA from pmApple-C1 (*Kremers et al., 2009*) was PCR amplified with primers: 5'- CTTTAAGAAGGA GATATACATATGGTGAGCAAGGGCGAGG-3'; 5'- CCACCCGGCCATGGAAGCTTGAGC –3'. Fragments of mApple and NdeI-linearized pMW172 (*Eads et al., 1998*) DNA containing human PFN1 (NCBI Gene ID: 5216) were joined via Gibson assembly. For mammalian expression, the mAp-PFN1 cassette was synthesized and inserted into pmApple-C1 behind the CMV promoter at the NdeI and BamHI restriction sites (Genscript). The amino acid linker and PFN1 were synthesized into the backbone of a pcDNA-Halo expression vector (provided by Gunther Hollopeter, Cornell University) flanked by KpnI and NotI sites (Genscript). We did not test additional flexible linkers of varying amino acid length or composition. Site-directed mutagenesis introduced the Y6D, R88E, G118V mutations into Halo-PFN1 (Genscript). The final sequence of each plasmid was confirmed (Genewiz, South Plainfield, NJ).

## Protein purification

PFN1 and mAp-PFN1 in pMW172 were transformed and expressed in Rosetta2(DE3) pRare2 competent cells and purified as described in detail (*Liu et al., 2022*). Notably, mAp-PFN1 stored >1 year at –80°C occasionally contains aggregates. We recommend pre-clearing the protein via ultracentrifugation before use. N-terminally tagged 6 × His-GFP-thymosin-β4 (GFP-Tβ4) was synthesized and cloned into a modified pET23b vector at the AgeI and NotI restriction sites (Genscript). Bacteria were transformed, induced, collected, and stored identically to PFN1 (*Liu et al., 2022*). Cell pellets were resuspended in 2 × PBS (pH 8.0) (2.8 M NaCl, 50 mM KCl, 200 mM sodium dibasic, 35 mM potassium monobasic), 20 mM imidazole (pH 7.4), 500 mM NaCl, 0.1% Triton-X 100, 14 mM BME and lysed. Lysate was clarified via centrifugation for 30 min at $20,000 \times g$ and the supernatant was flowed over cobalt affinity columns (Cytiva) equilibrated in low imidazole buffer (1 × PBS; (pH 8.0), 20 mM imidazole; (pH 7.4), 500 mM NaCl, 0.1% Triton-X 100, and 14 mM BME). GFP-Tβ4 was eluted using a linear gradient into high imidazole buffer (1 × PBS (pH 8.0), 300 mM imidazole (pH 7.4), 150 mM NaCl, 0.1% Triton-X 100, and 14 mM BME). The 6 × His tag was cleaved with 5 mg/mL ULP1 protease for 2 hr at 4°C, concentrated, and applied to a Superdex 75 (10/300) gel filtration column equilibrated in 1×PBS (pH 8.0), 150 mM NaCl, and 14 mM BME. Fractions containing GFP-Tβ4 were pooled, aliquoted, and stored at –80°C. The constitutively active formin 6×His-mDia1(FH1-C) (amino acids 571–1255) was synthesized and cloned into modified pET23b vector at the AgeI and NotI restriction sites (Genscript). Purification of 6 × His-mDia1(FH1-C) was handled exactly as described for GFP-Tβ4, but a Superose 6 Increase (10/300; Cytiva) column was used for gel filtration.

Rabbit skeletal muscle actin (RMA), OG, and N-(1-pyrenyl)iodoacetamide (pyrene) actin were purified from acetone powder and labeled on cysteine 374 as described (*Spudich and Watt, 1971*; *Cooper et al., 1984*; *Kuhn and Pollard, 2005*). Alexa-647 (NHS ester) actin was labeled as above but on lysine residues (*Hertzog and Carlier, 2005*). OG- and Alexa-647 actin were stored at –20°C in G-buffer with 50% glycerol. Unlabeled, biotinylated, and pyrene-labeled actins were flash frozen in G-buffer and stored at –80°C. Glycerol stored actins were dialyzed in G-buffer at 4°C overnight and pre-cleared via ultracentrifugation at $279,000 \times g$ before use. Tubulin was purified from *Bovine* brains by three cycles of temperature-induced polymerization and depolymerization (*Castoldi and Popov, 2003*). Labeled tubulin was purchased from Cytoskeleton, Inc (Denver, CO). AlexaFluor-647 GMP-CPP microtubule seeds were made as in *Groen et al., 2014*. Unlabeled tubulin was recycled before use.

All protein concentrations were determined by band densitometry from Coomassie-stained SDS-PAGE gels, compared to a BSA standard curve. Stoichiometries were determined using spectroscopy, extinction coefficients, and correction factors: actin $\varepsilon_{290} = 25,974$ $M^{-1}$ $cm^{-1}$, pyrene $\varepsilon_{339} = 26,000$ $M^{-1}$ $cm^{-1}$, OG $\varepsilon_{496} = 70,000$ $M^{-1}$ $cm^{-1}$, Alexa-647 $\varepsilon_{650} = 239,000$ $M^{-1}$ $cm^{-1}$. The correction factor for OG was 0.12, and 0.03 for Alexa-647.

## Liposome pelleting assays

Liposomes 55% 16:0 PC, 25% 16:0 PS, 18% 16:0 PE, and 5% 18:1 (0.33 mM final) PI phosphates ($PI(3,5)P_2$ or $PI(4,5)P_2$; Avanti Polar lipids, Alabaster, AL) were prepared according to *Banerjee and Kane, 2017*. Liposomes were resuspended in 9:20:1 $CH_3OH$:$CHCl_3$:$H_2O$, lyophilized for 30–40 min at 35°C, rehydrated in ice-cold 25 mM NaCl (pH 7.4) and 50 mM Tris-HCl, freeze-thawed five times, and extruded through a 100 nm filter twenty times. Liposomes and 1 µM PFN1 or mAp-PFN1 were

incubated for 30 min at room temperature, and pelleted at 400,000 × $g$. Collected supernatants and pellets were resuspended in equal volumes (100 µL) of buffer, precipitated in 10%$_{(v/v)}$ trichloroacetic acid, washed with cold 100% acetone, and dissolved in 50 µL of 50 mM Tris-HCl (pH 6.8), 8 M urea, 5% SDS, and 1 mM EDTA. Blots were probed as described in figure legends. Densitometry was performed in Fiji (*Schindelin et al., 2012*).

## Fluorescence polarization binding assays

Reactions with actin (10 nM; unlabeled or OG-labeled) were performed in 1 × PBS (pH 8.0) and 150 mM NaCl and incubated at room temperature for 15 min before recording. Competitive experiments had 10 nM unlabeled actin, 10 nM GFP-Tβ4, and concentrations of either PFN1 protein. Direct tubulin-binding experiments were performed with 10 nM tubulin in 1 × BRB80 (80 mM PIPES, 1 mM MgCl$_2$, and 1 mM EGTA; pH 6.8) with 150 mM NaCl, and concentrations of mAp-PFN1, at 4°C. No microtubules were found when the contents of the highest tubulin-containing reaction were spotted onto coverslips and screened for microtubules by epifluorescence microscopy (561 filter) at the end of the experiment. Time-lapse polarization was performed to determine the rate of nucleotide exchange on actin using 2 µM actin (unlabeled), 500 nM ATP-ATTO-488, and 1 µM PFN1 or mAp-PFN1 in NFG (5 mM Tris (pH 8), 0.2 mM CaCl$_2$, 0.5 mM DTT) supplemented with MEI (1 mM MgCl$_2$, 1 mM EGTA, and 10 mM imidazole-HCl (pH 7.0)), similar to *Colombo et al., 2021*. Polarization was recorded every 7.5–8 s for 30 min. Actin was dialyzed for 30 min in NFG and pre-cleared before use. Polarization of OG, GFP or ATTO-488 probes was determined at 440 nm/510 nm, whereas assays using the mAp-PFN1 were set to 568 nm/592 nm. Assays were performed in a plate reader (Tecan, Männedorf, Switzerland). All proteins and reagents were pre-cleared via centrifugation at 279,000×$g$. Non-linear curve fits were performed using data normalized so that the smallest mean in each data set was defined as zero. Data were fit to the following curve using least squares regression with no constraints: $Y = Y_0 - B_{max}*(X/(K_D + X))$.

## Bulk actin assembly assays

Bulk assembly assays were performed by combining 2 µM Mg-ATP actin (5% pyrene labeled), proteins or control buffers, and initiation mix (2 mM MgCl$_2$, 0.5 mM ATP, 50 mM KCl). Reactions for each replicate were initiated simultaneously by adding actin to reactions using a multichannel pipette. Total fluorescence was monitored at 365 nm/407 nm in a plate reader. Recorded values were averaged between three replicates.

## In vitro TIRF microscopy assays

TIRF microscopy was executed as described in *Henty-Ridilla, 2022*, with the exception that actin reactions were performed in 20 mM imidazole (pH 7.4) 50 mM KCl, 1 mM MgCl$_2$, 1 mM EGTA, 0.2 mM ATP, 10 mM DTT, 40 mM glucose, and 0.25% methylcellulose (4000 cP). TIRF microscopy used a DMi8 inverted microscope equipped with 120–150 mW solid-state lasers, a 100 × Plan Apo 1.47 NA oil-immersion TIRF objective (Leica Microsystems, Wetzlar, Germany), and an iXon Life 897 EMCCD camera (Andor; Belfast, Northern Ireland). Frames were captured at 5 s intervals. Parameters of actin or microtubules dynamics were determined from TIRF images as in *Henty-Ridilla, 2022*.

## Mammalian cell culture and cell lines

N2a (Synthego, Menlo Park, CA), NIH 3T3 (ATCC, Manassas, VA), or U2OS (ATCC) cells were grown in DMEM supplemented with 200 mM L-glutamine, 10% FBS and 1 × penicillin (500 U/mL)-streptomycin (500 µg/mL). All cell lines were screened for mycoplasma at regular intervals, by screening fixed cells for irregular (non-nuclear associated) DAP1 stain. All transfections were performed using Lipofectamine 3000 according to manufacturer's instructions for 6-well plates with 75 K cells and 100–200 ng plasmid per well. Cells were lysed (for blots) or imaged 18–24 hr following transfection. To generate clonal lines, pooled PFN1 CRISPR cells (Synthego, Menlo Park, CA) were diluted to 0.5 cells per well, grown in conditioned media, and confirmed via blot. Blots were probed with anti-PFN1 (1:3500; SantaCruz 137235, clone B-10) and anti-α-tubulin (1:10,000; Abcam 18251) primary antibodies for 1 hr then washed with 1 × TBST (20 mM Tris (pH8.0), 150 mM NaCl, and 0.1% Tween-20), and probed with goat anti-mouse:IRDye 800CW (1:5000; LI-COR Biosciences 926–32210), and donkey anti-rabbit 926–68073 (1:20,000) antibodies for 1 hr and washed with 1 × TBS. Fluorescent secondary antibodies

were detected with a LI-COR Odyssey Fc (LI-COR Biosciences, Lincoln, NE) and quantified by densitometry in Fiji software. For cell proliferation comparisons, 100 K PFN1[(+/+)] wild-type or PFN1[(-/-)] knockout cells were seeded per T25 flask, passaged, and counted every 4 days.

### Determining the concentration of PFN1 in N2a cells

The concentration of PFN1, mAp-PFN1, or Halo-PFN1 was determined using quantitative Western blots from 100 K confluent cells lysed in 200 μL 2 × Laemmli buffer. The total mass of PFN1 was determined from 50 μL of lysate and determined from PFN1 or mAp-PFN1 standard curves. The mean cell volume of a typical N2a cell was calculated as 196 μm$^3$ ($1.96 \times 10^{-13}$ L) by taking the average XY area of ten well-spread N2a cells and then multiplying by the mean cell thickness in Z (~1 μm) from the same cells, similar to *Christ et al., 2010*; *Cadart et al., 2017*. Correction of lysates was not necessary as transfection efficiencies were 70–90% for these cells (*Henty-Ridilla et al., 2016*; *Henty-Ridilla et al., 2017*).

### Cell morphology imaging and measurements

The morphology of N2a cells was standardized by plating 200 K PFN1[(+/+)] or PFN1[(-/-)] cells on medium-sized fibronectin coated Y-pattern coverslips (CYTOO, Inc, Grenoble, France). After 1 hr unattached cells were aspirated. Remaining cells were washed in 0.3% glutaraldehyde and 0.25% Triton X-100 diluted in 1 × PBS and fixed in 2% glutaraldehyde for 8 min. Autofluorescence was quenched with freshly prepared 0.1%$_{(w/v)}$ sodium borohydride. Coverslips were blocked for 1 h in 1% BSA$_{(w/v)}$ diluted in 1 × PBST (1 × PBS and 0.1% Tween-20), incubated with anti-α-tubulin (1:250; Abcam 18251) primary antibody resuspended in 1 × PBST for 1 hr, washed three times in 1 × PBST and probed with AlexaFluor-568 donkey anti-rabbit secondary antibody (A10042) and AlexaFluor-488 phalloidin (1:500; Life Technologies, Carlsbad, CA) to stain actin filaments. After 1 hr, coverslips were washed and mounted in AquaMount.

Cells were imaged by spinning disk confocal microscopy on an inverted Nikon Ti2-E microscope (SoRa; Nikon Instruments, Melville, NY) equipped with 488 nm, and 561 nm wavelength lasers, a CF160 Plan Apo 60 × 1.4 NA oil-immersion objective, a CSU-W1 imaging head (Yokogawa Instruments, Tokyo, Japan), a SoRa disk (Nikon Instruments, Melville, NY), and a Prime BSI sCMOS camera with a pixel size of 6.5 μm/pixel (Teledyne Photometrics, Tucson, AZ). Artificial intelligence denoise and 40 iterations of Richardson-Lucy deconvolution was applied to 7–10 μm Z-stacks acquired with identical laser exposures and power using Nikon Elements software. Images for cell morphology (saturated actin signal), actin fluorescence or microtubule fluorescence quantification were converted to 8-bit grayscale, binarized, and counted (RawIntDen) in Fiji software.

### Live-cell imaging

Cells were imaged via SoRa (as above) in phenol-red free DMEM (supplemented as above) buffered with 10 mM HEPES (pH 7.4) at 37°C. Temperature was maintained with a stage heater insert (OKO labs, Ambridge, PA). 42 K cells were plated in glass-bottom 35 mm dishes (MatTek, Ashland, MA) and transfected with 100 ng maxi-prepped plasmid per dish: reduced expression GFP-beta-actin (*Watanabe and Mitchison, 2002*), ensconsin-microtubule binding domain (EMTB)–2 × mCherry (*Miller and Bement, 2009*), and Halo-PFN1 (wild-type or mutants; this work). Janelia Fluor-646 (JF-646; Promega, Madison, WI) was added directly to dishes 5 min before acquisition.

### Data analyses and availability

GraphPad Prism 9 (GraphPad Software, San Diego, CA) was used for analyses and statistical tests. The design, sample size, and statistics used for each experiment are in each figure legend. All datasets passed tests for normality. Individual data points are shown in each figure shaded by replicate. Source datasets for all quantitative data are linked in the figure legends. The Henty-Ridilla Zenodo contains the original large image files and is available upon confirmation that users will follow CC-BY licensing guidelines for reuse at: http://doi.org/10.5281/zenodo.5329584.

## Acknowledgements

We are grateful to Marc Ridilla (Repair Biotechnologies) and Brian Haarer (SUNY Upstate) for helpful comments, and ASAPbio for a crowd sourced peer review. Research was funded by a Sinsheimer

Scholar Award, Amyotrophic Lateral Sclerosis Association Starter Grant (20-IIP-506), and the National Institutes of Health, GM133485.

## Additional information

### Funding

| Funder | Grant reference number | Author |
|---|---|---|
| Sinsheimer Foundation | Scholar Award | Jessica L Henty-Ridilla |
| ALS Association | 20-IIP-506 | Jessica L Henty-Ridilla |
| National Institutes of Health | GM133485 | Jessica L Henty-Ridilla |

The funders had no role in study design, data collection and interpretation, or the decision to submit the work for publication.

### Author contributions
Morgan L Pimm, Xinbei Liu, Farzana Tuli, Formal analysis, Investigation, Writing – review and editing; Jennifer Heritz, Ashley Lojko, Formal analysis; Jessica L Henty-Ridilla, Conceptualization, Formal analysis, Funding acquisition, Investigation, Methodology, Resources, Supervision, Visualization, Writing – original draft, Writing – review and editing

### Author ORCIDs
Jessica L Henty-Ridilla http://orcid.org/0000-0002-7203-8791

### Decision letter and Author response
Decision letter https://doi.org/10.7554/eLife.76485.sa1
Author response https://doi.org/10.7554/eLife.76485.sa2

## Additional files

### Supplementary files
• Transparent reporting form

### Data availability
Datasets for each figure have been uploaded. Original image files (very large size) have been deposited in the Zenodo Henty-Ridilla laboratory community, available here: https://doi.org/10.5281/zenodo.5329584. Readers/users should take note of and abide by the CC-BY license in the repository when reusing any of the data.

The following dataset was generated:

| Author(s) | Year | Dataset title | Dataset URL | Database and Identifier |
|---|---|---|---|---|
| Pimm ML | 2021 | files associated with HR_1 | https://doi.org/10.5281/zenodo.5329584 | Zenodo, 10.5281/zenodo.5329584 |

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
