## [Editor Report]

The development and rigorous characterization of fully functional, fluorescently labeled versions of profilin will be useful to cell biologists and biochemists who study the cytoskeleton. For cell biologists, these tools will allow a better understanding of the cellular dynamics of profilin and its interactions with different components of the cytoskeleton (actin networks and microtubules). Fluorescent profilins will also be precious to understand the consequences of different mutations. For biochemists, these tools offer new possibilities to study profilin dynamics in bulk assays or by live imaging.

---

## [Decision Letter]

**Decision letter after peer review:**

Thank you for submitting your article "Visualizing functional human profilin in cells and in vitro applications" for consideration by *eLife*. Your article has been reviewed by 2 peer reviewers, including Alphee Michelot as Reviewing Editor and Reviewer #1, and the evaluation has been overseen by Anna Akhmanova as the Senior Editor.

Essential revisions:

You will see that we easily reached a consensus as our reviews are quite similar. In general, we appreciated the good quality of your work. However, we also found that efforts should be made to better explain how your tools were developed, and why they will be useful for our community in the future. On this last point, it would be useful to discuss concrete examples of future applications. In addition, we found at times a lack of explanation of how the experiments were conducted. Please make an effort to clarify how each experiment was conducted, and ensure that each conclusion written in the text is clearly supported by experiments presented in this work.

Finally, a few additional experiments are suggested by the reviewers, which I believe should be feasible without much difficulty.

*Reviewer #1 (Recommendations for the authors):*

This work is of great quality but deserves a bit of extra work to correct few issues. Please find below are my suggestions for corrections and improvements.

1. To complete the story and confirm the normal interaction with G-actin, the authors could easily confirm that labeled profilin accelerates nucleotide exchange as well as native profilin.

2. The experiment of Figure S2 is not exactly showing that untagged profilin is unsuitable for conventional anisotropy assays. The reason why anisotropy signals do not change in this experiment is that the fluorescence lifetime of Oregon green is too short for proteins of the size of actin. But I am pretty sure that such experiment would work with longer lifetime fluorophores. KU dyes, for example, sells quite good ones.

3. Could the author clarify whether Oregon-Green Actin is labeled on lysines or Cys374? I could not find this information which is important to analyze the data. If labeling is on lysines, I am fine with it. If labeled on Cys374, how do the authors analyze their data as binding to profilin should be compromised?

4. TIRF movies give the impression of labeled-profilin spots of different sizes, which are often signs of the presence of aggregates. Is this true? Is this due to the TIRF buffer or to the presence of G-actin? What is the gel filtration profile of an mApple-profilin preps?

The experiment showing the binding of labeled profilin to the barbed ends of actin filaments seems nice and must be difficult to perform. I was wondering if the authors tried to lower the amount of G-actin. With actin filaments elongating more slowly, perhaps such events would be easier to detect. What bothers me here, at 1µM of G-actin, is that each filament should elongate by about 50 actin subunits in the 5 seconds they detect profilin binding. Therefore, detection at barbed ends should theoretically be impossible at this recording time interval, which makes me wonder if profilin binding in this video is real or just a coincidence.

5. There are things I don't understand in the experiments of Figure 7A. First of all, the authors talk about experiments performed at the "recommended dilution of JF-646 ligand", but I could not find what concentration this is. I assume from reading what follows that it is 2 µM, 1µM or 100 nM, but I am not sure.

The authors then evaluate that only 2% of Halo-profilin is labeled at this "recommended" concentration. How do they evaluate that? If this is the case, I would expect that further dilution of the JF-646 ligand would result in a decrease in fluorescence intensity but that the overall fluorescence profile would be conserved (unless the dilution was such that single molecules began to be seen which does not seem to be the case). Why isn't it the case? Isn't that rather an indication that a high fraction of JF-646 remains unbound to Halo-profilin above 100 nM?

At nanomolar concentration, the authors write that Halo-profilin appears localized to the cytoplasm, but I only see clusters in their images (the image of 7A-10 nM does not look very different to me from the image 7B-G118V where the authors talk about aggregates). What are these aggregates in 7A-10 nM?

The authors mention in the introduction that profilin has been detected on stress fibers. Do the authors have any idea why they can't recapitulate this observation?

6. Any idea why only 60% of the cells show localization of profilin to microtubules?

7. I find that some parts of the discussion repeat much of what is already mentioned in the introduction and Results sections. These parts of the discussion could be removed without affecting the quality of the manuscript. On the other hand, I found that for a manuscript describing new tools, the authors did not take advantage of the discussion to describe potential future applications. This would make the discussion even more exciting and would help less specialized readers see the value of these tools.

*Reviewer #3 (Recommendations for the authors):*

In this manuscript, Pimm and colleagues report their development of a functional probe that enables in vivo visualization of profilin, a cytoskeletal protein that interacts with both actin and microtubules. Because of its dual impacts on actin and microtubule dynamics, profilin is a critical regulatory hub for cytoskeletal assembly and cellular morphology. Through systematic characterization, the authors find that attachment of a fluorescent protein (or Halo tag) and 10 amino acid linker at the N-terminus of human profilin-1 does not decrease profilin's lipid-, actin-, or poly-L-proline-binding functions, or its ability to influence actin and microtubule assembly. Further, expression of Halo-tagged profilin-1 rescues neuroblastoma cells from profilin-1 knockout-induced morphological and cytoskeletal defects. Finally, the localization of Halo-tagged profilin-1 to microtubule networks in these cells requires an intact tubulin-binding interface but is unaffected by disruption of its actin- or poly-L-proline functions.

The experiments are well designed, and the data are robust. My suggestions are aimed at clarifying a few questions I had while reading the manuscript. They are included below, in the approximate order in which they appear in the text:

(1) On line 58, the authors write: "Some profilin outcompetes actin bound to intracellular lipids…". This sounds very interesting, and I would like to make certain that I understand what this means. Does profilin displace actin that is bound to intracellular lipids?

(2) How was the length of the linker between mApple and profilin selected? I assume that the previous studies that placed a GFP-derived fluorescent protein at profilin's N-terminus (referred to on line 82) did not use such a linker. Do the authors have insight into how sensitive profilin's functions are to the linker length?

(3) The authors state on line 162 that their assays with liposomes demonstrate that profilin and tagged-profilin bind PI(3,5)P2 or PI(4,5)P2 with similar affinity. Have the authors measured this affinity? If not, this interpretation might be a bit too strong, because it is difficult to compare affinities when only a single concentration is assayed. For example, if the lipid binding sites are saturated at this concentration, a change in the binding affinity would not necessarily result in a change in the "fraction bound". I would suggest rewording the sentence to emphasize that the fluorescent tag does not disrupt (or preclude) binding of profilin to the two PIP lipids.

(4) On lines 261-263, I think the authors may have meant to say that the TIRF reactions containing profilin had significantly less fluorescence than the control reaction that contained only actin and mDia1. Also, the text states that these reactions were carried out using OG-labeled actin, whereas the figure legend states that the actin was labeled with Alexa 647. My understanding from the Methods section is that the Alexa 647-labeled actin is labeled on lysines, whereas OG-labeled actin is labeled on cysteines. Since the position of the label on the actin has implications for the fluorescence of filaments assembled by formins, this point should be clarified.

(5) On line 273, the authors state that "in the presence of mDia1, mApple-profilin stimulates actin filament assembly in a manner similar to the untagged protein". Do the authors mean that both profilins stimulate filament elongation to the same extent? I ask because Figures 4C and 4D reveal that both profilins decrease nucleation compared to what is observed in reaction containing only actin and mDia1.

(6) On line 371, the cell morphology index is defined as "the ratio of endogenous cell area to other cell conditions for cells plated on micropatterns of the same size". However, in the legend for Figure S6H, this measurement is described as "the ratio of cell area to endogenous control cells". It appears that one of these descriptions might be the inverse of the other. Can the authors specify which description is correct?

(7) Does the morphology index for actin and microtubules report the fraction of the cell area that is occupied by polymerized actin or microtubules? Is this value normalized relative to the cell size or fluorescence intensity (i.e., density of the cytoskeletal network)?

(8) Does the fraction of cells in which profilin associates with microtubules differ between wild-type and Halo-Y6D profilin-expressing cells? The authors state that "Halo-Y6D shifts profilin toward microtubules" (line 516), so I am unsure how to interpret the quantification in Figure 7C. Is the difference between the PFN (WT) and Y6D measurements shown in this graph significant?

(9) The effect of profilin on microtubule elongation and stability is striking. Do profilin knockout cells expressing Halo-G118V have altered microtubule morphologies relative to cells in which Halo-profilin-1 is expressed?

---

## [Author Response]

Essential revisions:You will see that we easily reached a consensus as our reviews are quite similar. In general, we appreciated the good quality of your work. However, we also found that efforts should be made to better explain how your tools were developed, and why they will be useful for our community in the future. On this last point, it would be useful to discuss concrete examples of future applications. In addition, we found at times a lack of explanation of how the experiments were conducted. Please make an effort to clarify how each experiment was conducted, and ensure that each conclusion written in the text is clearly supported by experiments presented in this work.Finally, a few additional experiments are suggested by the reviewers, which I believe should be feasible without much difficulty.

Thank you for this positive summary. We are sorry for the delay returning these very reasonable revisions. We have performed all suggested experiments. We have also worked to remove unnecessary redundancy and increase overall clarity in text.

Reviewer #1 (Recommendations for the authors):This work is of great quality but deserves a bit of extra work to correct few issues. Please find below are my suggestions for corrections and improvements.

Thank you for the compliment and your very helpful suggestions. We address your concerns point-by-point below.

1. To complete the story and confirm the normal interaction with G-actin, the authors could easily confirm that labeled profilin accelerates nucleotide exchange as well as native profilin.

We now confirm that labeled profilin (mAp-PFN1) accelerates nucleotide exchange on actin monomers using ATTO-488-ATP and fluorescence polarization (Figure 3C). Notably, mAp-PFN1 and tag-free PFN1 performed at similar levels, elevated compared to actin-alone control.

2. The experiment of Figure S2 is not exactly showing that untagged profilin is unsuitable for conventional anisotropy assays. The reason why anisotropy signals do not change in this experiment is that the fluorescence lifetime of Oregon green is too short for proteins of the size of actin. But I am pretty sure that such experiment would work with longer lifetime fluorophores. KU dyes, for example, sells quite good ones.

Thank you for this information!!! We based our original text off information in Vinson et al. (1998), but we did not consider the different lifetimes for various “green” fluorophores until performing the requested experiment above. We have kept this data, but modified the text to more accurately reflect these considerations.

3. Could the author clarify whether Oregon-Green Actin is labeled on lysines or Cys374? I could not find this information which is important to analyze the data. If labeling is on lysines, I am fine with it. If labeled on Cys374, how do the authors analyze their data as binding to profilin should be compromised?

The OG-actin and pyrene actin used throughout this manuscript is labeled on Cys374. Whereas, the Alexa-647 actin is labeled on general lysine residues. This important labeling information, is now more clearly articulated in the methods. All direct binding assays (except for the one mentioned above, which we didn’t end up pursuing anyway) were performed with unlabeled actin and GFP-TB4 or mAp-PFN1 as the labeled moiety.

TIRF assays in Figure 3 were performed with OG-actin, whereas the TIRF assays in Figure 4 were performed in the lysine-labeled actin. This data confirms the reviewer’s concern; thus Figure 3 and Figure 4 experiments should not be compared to each other since they use different actins. We include similar controls with each set of experiments to account for this (i.e., actin alone control, actin and PFN1 control, or actin and mAp-PFN1 control).

4. TIRF movies give the impression of labeled-profilin spots of different sizes, which are often signs of the presence of aggregates. Is this true? Is this due to the TIRF buffer or to the presence of G-actin? What is the gel filtration profile of an mApple-profilin preps?

Great eyes! We are not certain why the spots are of different sizes, except that it may be correlated with slide coating quality, or as the reviewer points out, aggregates. We include the gel filtration trace from one of the two purifications of mAp-PFN1 (both were similar) in Figure 1 Supplement 1. We highlight the volumes where mAp-PFN1 present in the volumes eluted from the gel filtration column. Following concentration, the pooled protein does not appear degraded on a standard SDS-PAGE gel (Figure 1D). Older aliquots of the protein stored at -80 ^o^C (>1 year) occasionally have small pellets following clarification before use. We now emphasize this point in the methods.

The experiment showing the binding of labeled profilin to the barbed ends of actin filaments seems nice and must be difficult to perform. I was wondering if the authors tried to lower the amount of G-actin. With actin filaments elongating more slowly, perhaps such events would be easier to detect. What bothers me here, at 1µM of G-actin, is that each filament should elongate by about 50 actin subunits in the 5 seconds they detect profilin binding. Therefore, detection at barbed ends should theoretically be impossible at this recording time interval, which makes me wonder if profilin binding in this video is real or just a coincidence.

This was a great suggestion and we are grateful to the reviewer for pointing it out. When performing the assay with 0.5 µM actin and 5 µM mAp-PFN1 we observed more barbed-end colocalization events (Figure 3 Supplement 2C,D)! We are very excited about this but approaching it with an abundance of caution. The events that we observed still seemed very transient. It is possible with more optimization and a faster image detection system might make such observations more reliable/interpretable.

5. There are things I don't understand in the experiments of Figure 7A. First of all, the authors talk about experiments performed at the "recommended dilution of JF-646 ligand", but I could not find what concentration this is. I assume from reading what follows that it is 2 µM, 1µM or 100 nM, but I am not sure.The authors then evaluate that only 2% of Halo-profilin is labeled at this "recommended" concentration. How do they evaluate that? If this is the case, I would expect that further dilution of the JF-646 ligand would result in a decrease in fluorescence intensity but that the overall fluorescence profile would be conserved (unless the dilution was such that single molecules began to be seen which does not seem to be the case). Why isn't it the case? Isn't that rather an indication that a high fraction of JF-646 remains unbound to Halo-profilin above 100 nM?

For clarity, we now put the final concentration of JF-646 ligand present in each dish and leave the manufacturer’s suggestions out of it. The back of the envelope calculation we were trying to use to emphasize that only a subset of the total profilin present was being illuminated was confusing. We have removed it and now use plain language to make this point. At the lower concentrations of JF-646 ligand it may be single-molecules of profilin, but because of PFN1’s transient interactions with actin monomers are below the resolution of the scope it is really difficult to say for certain.

At nanomolar concentration, the authors write that Halo-profilin appears localized to the cytoplasm, but I only see clusters in their images (the image of 7A-10 nM does not look very different to me from the image 7B-G118V where the authors talk about aggregates). What are these aggregates in 7A-10 nM?

The authors are not certain that aggregates are present in Figure 8A because these molecules seemed quite dynamic while acquiring imaging stacks, whereas the molecules in 8B moved less. We did not perform (or currently have access to) FRAP to test whether these accumulations are truly aggregates or not. We have softened our wording around this in the text.

The authors mention in the introduction that profilin has been detected on stress fibers. Do the authors have any idea why they can't recapitulate this observation?

The original citation that observed PFN1 on stress fibers was with B16-F10 cells, which produce amazing stress fibers. We do not have those cells in the lab, and we rarely see stress fibers in the N2a cells used here (they make beautiful filopodia though). To assess this question, we expressed the Halo-PFN1 over endogenous PFN1 in two additional cell types (NIH3T3 and U2OS cells) and scored cells for association with F-actin or microtubules or both (Figure 8 Supplement 1). U2OS cells were very similar to N2a cells in that a substantial portion of cells displayed Halo-PFN1 association with microtubules with 10 nM JF-646 ligand. However, NIH3T3s displayed much more localization with actin filaments than microtubules. Differences in cell type explain this result. A more complicated explanation might be a difference in the amount of actin, tubulin, profilin in these cells.

6. Any idea why only 60% of the cells show localization of profilin to microtubules?

We were surprised to see 60% localization with microtubules! The only ideas we have are that there is more profilin free from binding actin monomers in a cell than previously thought or this is a cell specific measurement suggested from the requested experiment above (Figure 8C and Figure 8 Supplement 1).

7. I find that some parts of the discussion repeat much of what is already mentioned in the introduction and Results sections. These parts of the discussion could be removed without affecting the quality of the manuscript. On the other hand, I found that for a manuscript describing new tools, the authors did not take advantage of the discussion to describe potential future applications. This would make the discussion even more exciting and would help less specialized readers see the value of these tools.

We have substantially cut areas of repetition in the text. We have also focused the discussion on describing the potential uses of these tools in vitro and in cells.

Reviewer #3 (Recommendations for the authors):In this manuscript, Pimm and colleagues report their development of a functional probe that enables in vivo visualization of profilin, a cytoskeletal protein that interacts with both actin and microtubules. Because of its dual impacts on actin and microtubule dynamics, profilin is a critical regulatory hub for cytoskeletal assembly and cellular morphology. Through systematic characterization, the authors find that attachment of a fluorescent protein (or Halo tag) and 10 amino acid linker at the N-terminus of human profilin-1 does not decrease profilin's lipid-, actin-, or poly-L-proline-binding functions, or its ability to influence actin and microtubule assembly. Further, expression of Halo-tagged profilin-1 rescues neuroblastoma cells from profilin-1 knockout-induced morphological and cytoskeletal defects. Finally, the localization of Halo-tagged profilin-1 to microtubule networks in these cells requires an intact tubulin-binding interface but is unaffected by disruption of its actin- or poly-L-proline functions.The experiments are well designed, and the data are robust. My suggestions are aimed at clarifying a few questions I had while reading the manuscript. They are included below, in the approximate order in which they appear in the text:

Thank you for this very positive review. We found your comments extremely helpful and we are very grateful.

(1) On line 58, the authors write: "Some profilin outcompetes actin bound to intracellular lipids…". This sounds very interesting, and I would like to make certain that I understand what this means. Does profilin displace actin that is bound to intracellular lipids?

We have since reworded this for the sake of brevity and clarity. However, to our knowledge there is indeed a complex dynamic between profilin and certain PIP lipids including that profilin can displace actin bound to intracellular lipids.

(2) How was the length of the linker between mApple and profilin selected? I assume that the previous studies that placed a GFP-derived fluorescent protein at profilin's N-terminus (referred to on line 82) did not use such a linker. Do the authors have insight into how sensitive profilin's functions are to the linker length?

Thank you for this comment. Yes, we the previous GFP-derived PFN1 used in that citation did not have a flexible linker. We did not try other linker lengths and got incredibly lucky following the linker used by Michael Davidson’s lab with many other cytoskeleton protein fusions. Thus, we do not know with certainty if the tool could be further improved, but the tools seem robust for the characteristics assessed here. We now mention this in the text to help future users of these tools or curious readers of this manuscript.

(3) The authors state on line 162 that their assays with liposomes demonstrate that profilin and tagged-profilin bind PI(3,5)P2 or PI(4,5)P2 with similar affinity. Have the authors measured this affinity? If not, this interpretation might be a bit too strong, because it is difficult to compare affinities when only a single concentration is assayed. For example, if the lipid binding sites are saturated at this concentration, a change in the binding affinity would not necessarily result in a change in the "fraction bound". I would suggest rewording the sentence to emphasize that the fluorescent tag does not disrupt (or preclude) binding of profilin to the two PIP lipids.

This reviewer is 100% correct and thank you for pointing this out. We have removed claims of similar affinity for the liposome assays and now state that the tagged profilin is capable of binding rather than making any claims that the binding is equal since we did not try other concentrations.

(4) On lines 261-263, I think the authors may have meant to say that the TIRF reactions containing profilin had significantly less fluorescence than the control reaction that contained only actin and mDia1. Also, the text states that these reactions were carried out using OG-labeled actin, whereas the figure legend states that the actin was labeled with Alexa 647. My understanding from the Methods section is that the Alexa 647-labeled actin is labeled on lysines, whereas OG-labeled actin is labeled on cysteines. Since the position of the label on the actin has implications for the fluorescence of filaments assembled by formins, this point should be clarified.

As both reviewers have pointed out the way actin is labeled is critical for interpreting these results. Your understanding is correct, but we clarified which actin was used in each experiment to make this easier to decipher. We have reworded TIRF results for Figure 4B-C as suggested. The figure legend now accurately states that Alexa-647 actin was used for these experiments.

(5) On line 273, the authors state that "in the presence of mDia1, mApple-profilin stimulates actin filament assembly in a manner similar to the untagged protein". Do the authors mean that both profilins stimulate filament elongation to the same extent? I ask because Figures 4C and 4D reveal that both profilins decrease nucleation compared to what is observed in reaction containing only actin and mDia1.

Yes, we mean that in the presence of formin both profilins stimulate formin-based elongation to the same extent. We now clarify this in the text.

(6) On line 371, the cell morphology index is defined as "the ratio of endogenous cell area to other cell conditions for cells plated on micropatterns of the same size". However, in the legend for Figure S6H, this measurement is described as "the ratio of cell area to endogenous control cells". It appears that one of these descriptions might be the inverse of the other. Can the authors specify which description is correct?

We have clarified this in the text. We have reanalyzed this data to present it in (we hope) a more straightforward way (see reviewer point 7, below).

(7) Does the morphology index for actin and microtubules report the fraction of the cell area that is occupied by polymerized actin or microtubules? Is this value normalized relative to the cell size or fluorescence intensity (i.e., density of the cytoskeletal network)?

We have reanalyzed clarified these parameters in the text and in Review 1’s comment above. Basically, the only normalization applied is from plating cells on micropatterns. The rest of the analysis is based on thresholding (the same for each treatment but different for each parameter) and then a count of total binarized pixels present. These counts are represented as individual dots in Figures 6 and Figure 7, which were used to generate averages in the analysis. Cells were imaged from different coverslips and plated on different days.

(8) Does the fraction of cells in which profilin associates with microtubules differ between wild-type and Halo-Y6D profilin-expressing cells? The authors state that "Halo-Y6D shifts profilin toward microtubules" (line 516), so I am unsure how to interpret the quantification in Figure 7C. Is the difference between the PFN (WT) and Y6D measurements shown in this graph significant?

The differences in the graph (now in Figure 8C) were not significantly different. We have softened this claim in the text.

(9) The effect of profilin on microtubule elongation and stability is striking. Do profilin knockout cells expressing Halo-G118V have altered microtubule morphologies relative to cells in which Halo-profilin-1 is expressed?

This is a great suggestion and area we plan to use the tools described herein in the future. We now include these effects for cell morphology, and total actin filament and microtubule pixel counts (Figure 7). G118V expressing cells do not rescue PFN1-deficient cells for overall morphology, or the architecture of either actin filaments or microtubules. This may not be too surprising since G118V does not bind actin monomers or microtubules as well as wild-type profilin (Henty-Ridilla et al., 2017 and Liu et al. 2022).